# Reconstructing the Evolution of Ice Sheets, Sea Level and Atmospheric CO₂ During the Past 3.6 Million Years

Constantijn J. Berends[1], Bas de Boer[2], Roderik S. W. van de Wal[1,3]

[1] Institute for Marine and Atmospheric research Utrecht, Utrecht University, Utrecht, The Netherlands
[2] Earth and Climate Cluster, Faculty of Science, Vrije Universiteit Amsterdam, The Netherlands
[3] Faculty of Geosciences, Department of Physical Geography, Utrecht University, Utrecht, The Netherlands

*Correspondence to*: Constantijn J. Berends (c.j.berends@uu.nl)

**Abstract.** Understanding the evolution of, and the interactions between, ice sheets and the global climate over geological time scales is important for being able to project their future evolution. However, direct observational evidence of past $CO_2$
concentrations, and the implied radiative forcing, only exists for the past 800,000 years. Records of benthic $\delta^{18}O$ date back millions of years, but contain signals from both land ice volume and ocean temperature. In recent years, inverse forward modelling has been developed as a method to disentangle these two signals, resulting in mutually consistent reconstructions of ice volume, temperature and $CO_2$. We use this approach to force a hybrid ice-sheet – climate model with a benthic $\delta^{18}O$ stack, reconstructing the evolution of the ice sheets, global mean sea level and atmospheric $CO_2$ during the late Pliocene and
the Pleistocene, from 3.6 million years (Myr) ago to the present day. During the warmer-than-present climates of the Late Pliocene, reconstructed $CO_2$ varies widely, from 320 – 440 ppmv for warm periods, to 235 – 250 ppmv for the early glacial excursion ~3.3 million years ago. Sea level is relatively stable during this period, with maxima of 6 – 14 m, and minima of 12 – 26 m during glacial episodes. Both $CO_2$ and sea level are within the wide ranges of values covered by available proxy data for this period. Our results for the Pleistocene agree well with the ice-core $CO_2$ record, as well as with different available sea-
level proxy data. During the early Pleistocene, 2.6 – 1.2 Myr ago, we simulate 40 kyr glacial cycles, with interglacial $CO_2$ decreasing from 280 – 300 ppmv at the beginning of the Pleistocene, to 250 – 280 ppmv just before the Mid-Pleistocene Transition (MPT). Peak glacial $CO_2$ decreases from 220 – 250 ppmv to 205 – 225 ppmv during this period. After the MPT, when the glacial cycles change from 40 kyr to 80/120 kyr cyclicity, the glacial-interglacial contrast increases, with interglacial $CO_2$ varying between 250 – 320 ppmv, and peak glacial values decreasing to 170 – 210 ppmv.

**1 Introduction**

Understanding the response of ice sheets and the global climate as a whole to changes in the concentrations of atmospheric $CO_2$, is important for understanding the future evolution of the climate system. Since large-scale changes in ice sheet geometry typically occur over thousands to tens of thousands of years, sources of information other than direct observational evidence are required. In order to gain more insight in the relation between these components of the Earth system, studying their
evolution during the geological past is useful.

One particularly rich source of information is presented by ice cores. The chemical and isotopical content of the ice itself can provide valuable information on the state of the global climate, and in particular on the temperature, at the time the ice was formed (Dansgaard, 1964; Jouzel et al., 1997; Alley, 2000; Kindler et al., 2014). Air bubbles, trapped when the snow compresses first into firn and ultimately into ice, contain tiny samples of the atmosphere from the time the air got trapped. The oldest ice core presently available, the EPICA Dome C core, contains ice, and air bubbles and hydrates, dating back 800 kyr (Bereiter et al., 2015). The information obtained from these ice cores has greatly improved our understanding of the dynamics of the glacial cycles of the Late Pleistocene.

Different methods of relating chemical and isotopic properties of ocean sediments to the atmospheric $CO_2$ concentration have been used to create proxy data extending back further in time than the ice core record. Many studies have measured $\delta^{11}B$ of different fossil foraminifera, using the observed relation to seawater pH to calculate atmospheric $CO_2$ concentrations (Hönisch et al., 2009; Seki et al., 2010; Bartoli et al., 2011; Martínez-Botí et al., 2015; Foster and Rae, 2016; Stap et al., 2016; Chalk et al., 2017; Dyez et al., 2018; Sosdian et al., 2018). Another line of evidence has focused on the $\delta^{13}C$ of alkenones in fossil foraminifera (Seki et al., 2010; Badger et al., 2013; Zhang et al., 2013), although the reliability of this proxy for $CO_2$ concentrations lower than ~400 ppmv has recently been called into question (Badger et al., 2019). A different line of work relates the density of stomata on fossil plant leaves to atmospheric $CO_2$ concentration, providing data throughout the Cenozoic (Kürschner et al., 1996; Wagner et al., 2002; Finsinger and Wagner-Cremer, 2009; Beerling and Royer, 2011; Stults et al., 2011; Bai et al., 2015; Hu et al., 2015). This proxy has been the subject of discussion regarding its reliability (Indermühle et al., 1999; Jordan, 2011; Porter et al., 2019), based on its discrepancies with the ice-core record, and presently unresolved problems due to the influence of other effects such as evolution, extinction, changes in local environment, and immigration of species, although recent work has gone some way to resolving these issues (Franks et al., 2014).

Information about the global glacial state is furthermore obtained from the $\delta^{18}O$ of fossil benthic foraminifera, which is influenced by both total ice volume and deep-sea temperature. Ocean sediment cores containing foraminiferal shells have been used to create stacks of benthic $\delta^{18}O$ records dating back 65 Myr (Lisiecki and Raymo, 2005; Zachos et al., 2001, 2008). Fig. 1 shows the LR04 stack of benthic $\delta^{18}O$ records (Lisiecki and Raymo, 2005), together with the EPICA Dome C $CO_2$ record (Bereiter et al., 2015) and the Earth's orbital parameters and Northern hemisphere summer insolation (Laskar et al., 2004), during the last 800 kyr.

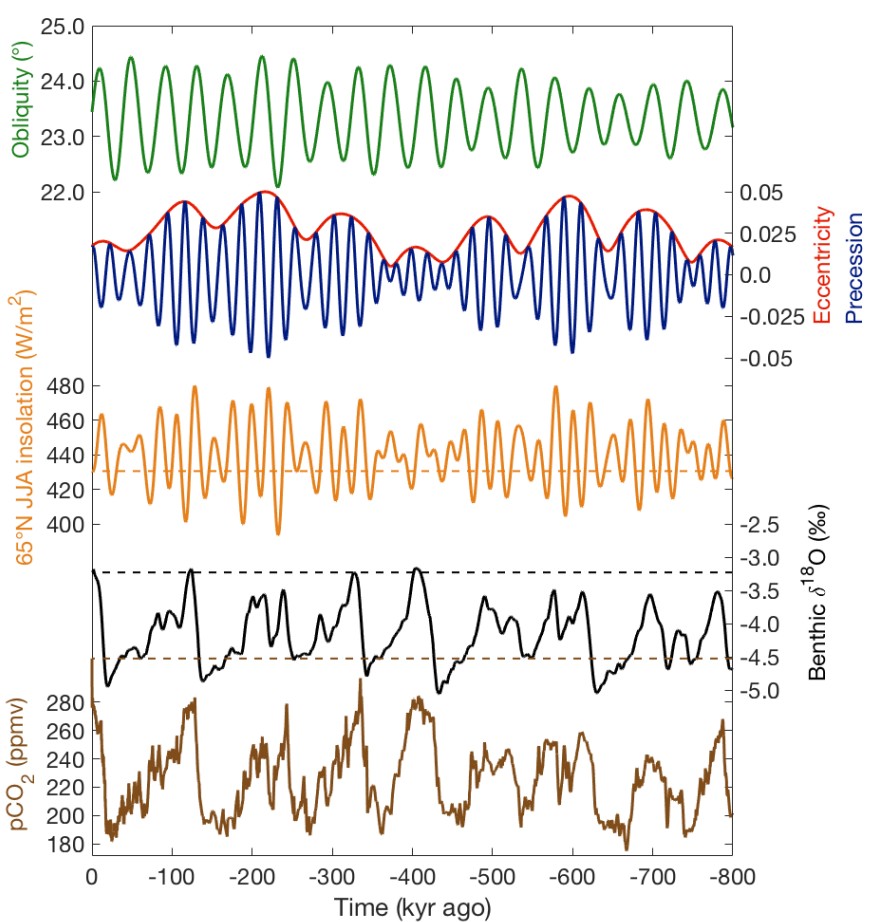

**Figure 1: Earth's orbital parameters (obliquity, eccentricity and precession) and the resulting Northern summer insolation (Laskar et al., 2004), benthic δ18O (Lisiecki and Raymo, 2005) and ice core CO₂ (Bereiter et al., 2015) during the past 800 kyr.**

Since benthic $\delta^{18}O$ contains both an ice-based and a climate-based signal, using it to understand the evolution of either of these two Earth system components is only possible after disentangling the two contributions. Several studies have aimed at performing this separation by deriving either of the two signals from independent other proxies. One approach has been to derive deep-water temperatures from foraminiferal Mg/Ca ratios (Sosdian et al., 2009; Elderfield et al., 2012; Shakun et al., 2015). Global mean sea level is then solved as a closure term. Alternatively, temperature proxies have been obtained from records of oxygen and hydrogen isotope abundances from ice cores from Greenland (Alley, 2000) and Antarctica (Jouzel et al., 2007). Focusing on the signal from ice volume rather than temperature, Rohling et al. (2014) used planktic $\delta^{18}O$ in the Mediterranean Sea as a proxy for sea-level at the Strait of Gibraltar, translating the result into a global mean sea-level record spanning the last 5.5 Myr. Grant et al. (2014) applied the same method to the Red Sea, producing a record of sea-level at the Bab-el-Mandab Strait with a higher accuracy, but going back only 500 kyr.

These studies are generally viewed as "data" studies; observed variables (chemical concentrations, isotope ratios, etc.) are used to derive climate parameters (ice volume, global mean temperature, $pCO_2$, etc.) using relatively simple, time-independent concepts. Discussions about the validity of the results generally focus less on the physical processes described by these simple models, and more on the properties of the data, such as sample contamination and diagenesis, statistical limitations and

measurement uncertainty.

A different family are the "model" studies, where physics-based models are used to describe the evolution of (components of) the Earth system (atmosphere, ocean, cryosphere, carbon cycle, etc.) through time. Typically, the aim of a model study is to determine if our understanding of a physical process is good enough to explain the observations, or to use that understanding to make predictions of that process in the future. Several recent studies have aimed to reproduce the evolution of global climate

and the cryosphere throughout the Pleistocene, using primarily insolation as forcing (Brovkin et al., 2012; Willeit et al., 2015, 2019). By studying the differences between model results and observations from proxy data, such studies can help with interpreting these data, providing insight into the physical processes that govern the relation between observed and derived variables. In turn, the differences between data and model results can help to assess the importance of physical processes that are not included in the model.

Inverse modelling is a hybrid method, using an approach that combines elements from both these families of research. This method aims to derive the evolution of the entire global climate-cryosphere system through time, based on observations of benthic $\delta^{18}O$ (Bintanja et al., 2005; Bintanja and van de Wal., 2008; de Boer et al., 2010, 2012, 2013, 2014, 2017; van de Wal et al., 2011; Stap et al., 2017, 2017; Berends et al., 2018, 2019). This is done by using ice-sheet models, with climate forcing components varying in complexity from simple parametrisations to fully coupled GCMs, to reconstruct ice-sheet evolution.

Such models can be used to calculate the contributions to the observed benthic $\delta^{18}O$ signal from both ice volume and deep-water temperature over time. By using a tool called an "inverse routine", the model can be forced to (almost) exactly reproduce the benthic $\delta^{18}O$ signal, providing a reconstruction of global climate *as it should have evolved* in order to explain the observations. The advantage of this approach over other proxy-based reconstructions is that the simulated changes in global climate and ice-sheet evolution are mutually consistent, building on the physical equations within the model framework. Earlier

studies adopting this approach used simple, one-dimensional ice-sheet models to represent all ice on Earth or on a single hemisphere, and different parameterisations of the relation between ice volume and climate (e.g. de Boer et al., 2010; Stap et al., 2017). More recent studies have used more elaborate 3-D ice-sheet models covering different regions of the Earth, using more comprehensive mass balance parameterisations and representations of the global climate (e.g. Bintanja and van de Wal, 2008; de Boer et al., 2013, 2017; Berends et al., 2018, 2019). The most recent of these studies, by Berends et al. (2019), used

a 3-D ice-sheet-shelf model, forced with output from several different GCM simulations by a so-called "matrix method" of model coupling, to simulate the last four glacial cycles. They showed that their results agreed with geomorphological and proxy-based evidence of ice-sheet volume and extent, benthic $\delta^{18}O$, deep-water temperature, ice-sheet temperature and atmospheric $CO_2$.

The work presented here builds on the work by Berends et al. (2019), extending their results to 3.6 Myr ago to produce a time-continuous, self-consistent reconstruction of atmospheric $CO_2$, global climate, ice-sheet geometry and global mean sea-level, over the period where ice core records of $CO_2$ are not available. The inverse modelling approach that was adopted to force this model with a benthic $\delta^{18}O$ record is described in Sect. 2.1. The ice-sheet and climate models are described in Sects. 2.2 and 2.3, respectively, while the matrix method used to couple these two models is described in Sect. 2.4. The resulting reconstructions of $CO_2$, ice volume, temperature and sea-level over the past 3.6 Myr are presented in Sect. 3 and discussed in Sect. 4.

## 2 Methodology

### 2.1 Inverse modelling

In order to disentangle the contributions to the benthic $\delta^{18}O$ signal from global land ice volume and deep-water temperature changes, we use the inverse modelling method proposed by Bintanja and van de Wal (2008) and refined by de Boer et al. (2013, 2014, 2017) and Berends et al. (2019). The two contributions to the benthic $\delta^{18}O$ signal are not independent; both result from changes in the Earth's climate, driven by changes in insolation and $CO_2$. A cooling of the global climate will, over time, affect the temperature in the deep ocean, especially when the cooling occurs at high latitudes, where deep water formation occurs. Simultaneously, such a global cooling leads to an increase in ice volume, which affects the $\delta^{18}O$ of the sea water itself. The inverse modelling method is a tool that determines how modelled $CO_2$ *should have evolved over time* in order to affect the global climate in such a way that the observed benthic $\delta^{18}O$ record is reproduced. The modelled value for $CO_2$ is used, together with the ice sheets simulated by the ice-sheet model, to determine the global climate by using a matrix method of model forcing. This method was presented by Berends et al. (2018) and refined by Berends et al. (2019), and is described in more detail in Sect. 2.4. The resulting climate is used to determine the surface mass balance over the ice sheets, which forces the ice-sheet model. The resulting global ice volume and deep-water temperature anomaly (which is derived from the high-latitude annual mean surface temperature anomaly, using a moving average of 3,000 yr) are used to calculate a modelled value of benthic $\delta^{18}O$. This approach is visualized conceptually in Fig. 2.

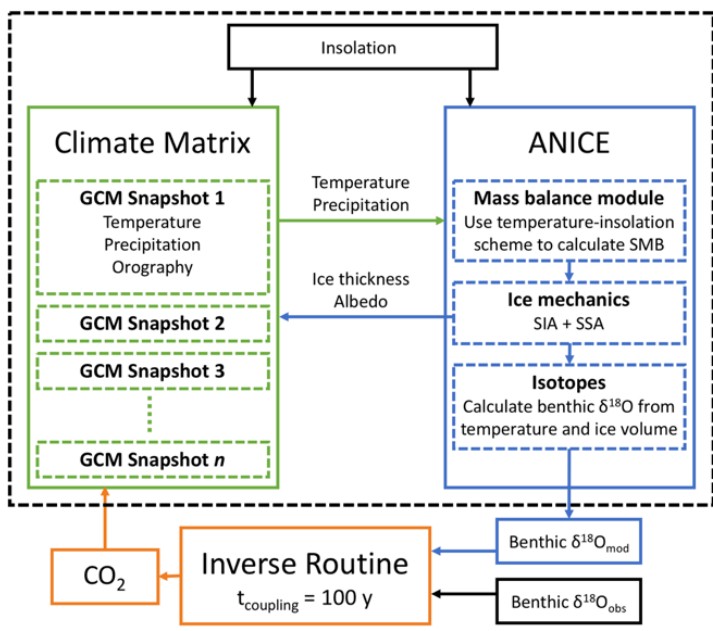

**Figure 2: A conceptual visualization of the inverse forward modelling approach. The model is forced externally by an insolation reconstruction and a benthic δ¹⁸O record (black boxes). pCO₂ is changed over time based on the difference between observed and modelled δ¹⁸O. The climate matrix interpolates between the pre-calculated GCM snapshots, based on the prescribed pCO₂ value and the modelled state of the cryosphere (ice thickness and albedo), to determine the climate that is used to calculate the surface mass balance over the ice sheets. The modelled ice sheets and climate are then used to calculate a modelled benthic δ¹⁸O value, which is used to update modelled CO₂ in the next time step. Figure adapted from Berends et al. (2019).**

The modelled value $\delta^{18}O_{mod}$ is compared to the observed value $\delta^{18}O_{obs}$ at the model time $t$. A modelled value that is too low indicates that the modelled climate is too warm, or the ice sheets are too small. pCO₂ is then decreased in the next time-step (with an increment proportional to the $\delta^{18}O$ discrepancy). This relationship is quantified by the following equation:

$$pCO_2 = \overline{pCO_2} + \alpha\left(\delta^{18}O_{mod} - \delta^{18}O_{obs}\right). \tag{1}$$

Here, $\overline{pCO_2}$ is the mean modelled pCO₂ over the preceding 8.5 kyr, and $\alpha$ is a scaling parameter which controls how fast modelled CO₂ is allowed to change (serving as a low-pass filter, suppressing overshoot that could result from large changes in the benthic $\delta^{18}O$ record). In order to calculate $\delta^{18}O_{mod}$, the contributions from modelled ice volume and deep-sea temperature are calculated separately. The spatially variable isotope content of the individual ice-sheets is tracked through time, with the surface isotope balance based on the observed present-day relation between precipitation rates and isotope content according to Zwally and Giovinetto (1997). Benthic $\delta^{18}O$ is assumed to be linearly dependent on the global mean deep-water temperature anomaly, which is calculated by temporally smoothing the high-latitude annual mean surface temperature anomaly over the preceding 3 kyr and multiplying with a scaling factor of 0.25. Berends et al. (2019) showed that this yields a deep-sea temperature anomaly of 2 – 2.5 K at LGM, in general agreement with proxy-based reconstructions (Shakun et al., 2015). The

values of 8.5 kyr for the length of the $CO_2$ averaging window, 3 kyr for the deep-water temperature averaging window, and 120 *ppmv ‰$^{-1}$* for the $\delta^{18}O$-$CO_2$ scaling parameter, were determined experimentally to accurately reproduce the observed benthic $\delta^{18}O$ record.

The combination of this inverse modelling method to reconstruct $pCO_2$ and the matrix method to determine the global climate has been shown to accurately reproduce changes in benthic $\delta^{18}O$ and the individual contributions from both global ice volume and deep-water temperature, as well as ice-sheet volume and extent, ice-sheet surface temperature and $pCO_2$ during the last four glacial cycles (Berends et al., 2019).

## 2.2 Ice-sheet model

The evolution of the ice sheets is simulated using ANICE, a coupled 3-D ice-sheet-shelf model (de Boer et al., 2013; Berends et al., 2018). ANICE uses a combination of the shallow ice approximation (SIA; Morland and Johnson, 1980) for grounded ice and the shallow shelf approximation (SSA; Morland, 1987) for floating ice to solve the ice mechanical equations. Basal sliding is described by a Coulomb sliding law and solved using the SSA, using the hybrid approach by Bueler and Brown (2009), where basal friction is determined by bedrock elevation. Internal ice temperatures, used to calculate ice viscosity, are calculated using a coupled thermodynamical module. The surface mass balance is parameterized based on monthly mean surface temperature and precipitation, where ablation is calculated using the surface temperature-albedo-insolation parameterization, as explained in more detail by Berends et al. (2018). The solution by Laskar et al. (2004) is used to prescribe time- and latitude-dependent insolation at the top of the atmosphere. A combination of the temperature-based formulation by Martin et al. (2011) and the glacial-interglacial parameterization by Pollard & DeConto (2009), is used to calculate sub-shelf melt underneath the Antarctic ice shelves, calibrated by de Boer et al. (2013) to produce realistic present-day Antarctic shelves and grounding lines. A more detailed explanation is provided by de Boer et al. (2013) and references therein. A simple threshold thickness of 200 m is used to describe ice calving, where any shelf ice below this threshold thickness is removed. The model is run on four separate grids simultaneously, covering North America, Eurasia, Greenland and Antarctica, as shown in Fig. 3. The horizontal resolution is 40 km for Antarctica, North America and Eurasia, and 20 km for Greenland.

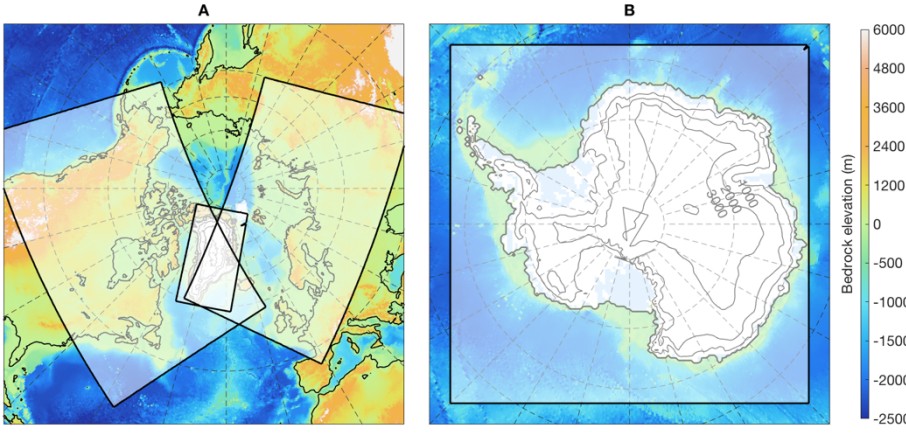

**Figure 3: The areas of the world covered by the four model domains of ANICE. In the North America and Eurasia domains, Greenland is omitted. Figure adapted from Berends et al. (2018).**

### 2.3 Climate model

5 HadCM3 is a coupled atmosphere-ocean GCM (Gordon et al., 2000; Valdes et al., 2017), which has been shown to accurately reproduce the present-day climate heat budget (Gordon et al., 2000). It has been used for future climate projections in the IPCC AR4 (Solomon et al., 2007), and paleoclimate reconstructions such as PlioMIP (Haywood and Valdes, 2003; Dolan et al., 2011, 2015; Haywood et al., 2013) and PMIP2 (Braconnot et al., 2007). Atmospheric circulation is calculated at a resolution of 2.5 ° latitude by 3.75 ° longitude. The ocean is modelled at a horizontal resolution of 1.25 ° by 1.25 °, with 20 vertical 10 layers. We use results from several different steady-state time slice simulations with HadCM3 of different climate states to force our ice-sheet model, using the matrix method explained in Sect. 2.4.

### 2.4 Matrix method

According to Pollard (2010), a climate matrix is a collection of pre-calculated output data from several steady-state GCM simulations, called "snapshots". These snapshots differ from each other in one or more key parameters, such as orbital 15 configuration, prescribed atmospheric pCO2, or ice-sheet configuration. Each of these constitutes a separate dimension of the matrix. When using an ice-sheet model to simulate the evolution of an ice sheet over time, the prescribed climate is determined in every model time-step by combining the GCM snapshots according to the position of the ice-sheet model state in the climate matrix, which is determined by the modelled values of the parameters describing the snapshots. This approach occupies the middle ground between offline forcing and fully coupled ice-sheet – climate models. The different GCM snapshots contain 20 the key feedback effects of the altitude and albedo on the temperature. In addition, the matrix method captures the effect of ice sheet geometry on large-scale atmospheric circulation and precipitation. The matrix method creates a spatially variable linear interpolation of these snapshots providing a first order approach to the strength of the feedback and the effects of ice-sheet geometry on atmospheric circulation and precipitation.

In this study, we use the matrix method developed by Berends et al. (2018), where temperature fields from the different climate states are combined based on a prescribed value for $pCO_2$ and on the internally modelled ice-sheets. The feedback of the ice sheets on the climate is calculated via the effect on absorbed insolation through changes in surface albedo. This interpolation is carried out separately for all four ice sheets. The altitude-temperature feedback is parameterized by a constant lapse-rate derived from the GCM snapshots. Precipitation fields are combined based on changes in surface elevation, reflecting the orographic forcing of precipitation and plateau desert effect caused by the presence of a large ice-sheet. The equations describing these calculations are presented in Appendix A, and explained in more detail by Berends et al. (2018), who demonstrated the viability of this method by simulating the evolution of the North American, Eurasian, Greenland and Antarctic ice-sheets throughout the entire last glacial cycle at the same time. They showed that model results agree well with available data in terms of ice-sheet extent, sea-level contribution, ice-sheet surface temperature and contribution to benthic $\delta^{18}O$. Here, we apply this matrix method to the climate matrix created by Berends et al. (2019), consisting of eleven pre-calculated GCM snapshots, created with HadCM3. Two of these, produced by Singarayer and Valdes (2010), respectively represent the pre-industrial period (PI) and the last glacial maximum (LGM). The other nine, produced by Dolan et al. (2015), represent the global climate during Marine Isotope Stage (MIS) M2 (3.3 My ago), for four different possible ice-sheet geometries and two different $pCO_2$ concentrations, plus one Pliocene control run. The total set of eleven snapshots allows the climate matrix to disentangle the effects on climate of changes in $pCO_2$ and changes in ice-sheet extent, and provides information on climates that are both colder and warmer than present-day.

## 3 Experimental set-up and results

Here, we describe a set of simulations from 3.6 Myr ago to the present day. The choice for this starting point aims to include the end of the Late Pliocene and the inception of the Pleistocene glacial cycles. The model was initialized with the same PRISM3 ice sheets that were used as boundary conditions in several of the HadCM3 simulations by Dolan et al. (2015). The model was forced with the LR04 stack of benthic $\delta^{18}O$ records (Lisiecki and Raymo, 2005). We chose here to perform three simulations: one "default" run, one with the $\delta^{18}O$ forcing adjusted upwards by 0.1 ‰ and one with the forcing adjusting downwards by 0.1 ‰. Berends et al. (2019) showed that the contribution to the uncertainty in the reconstructed $CO_2$ and sea level arising from the uncertainty in the forcing is the largest of all model parameters. Since simulating such long periods of time is very computationally demanding, we decided that the added value of repeating the sensitivity analysis for other model paramters by Berends et al., 2019 did not outweigh the computational cost. Margins of error reported here therefore describe only the uncertainty arising from the uncertainty in the benthic $\delta^{18}O$ record. The true uncertainty in our reconstruction, which includes contributions from many other parameterisations made in our model, is significantly larger, but remains difficult to quantify.

The resulting reconstructions of $CO_2$ and global mean sea-level, together with the $\delta^{18}O$ forcing, are shown in Fig. 4. The 40 kyr glacial cycles of the Early Pleistocene, between 2.6 and 1.2 Myr ago, are clearly visible, changing to 80/120 kyr cycles

after the MPT. The visibly larger uncertainty for higher CO₂ concentrations, occurring during the warm periods of the late Pliocene, is likely due to the relative sparsity of the climate matrix for warmer-than-present states (three snapshots only) compared to the colder-than-present part (seven snapshots), which might lead to a bias towards larger ice sheets in warm worlds.

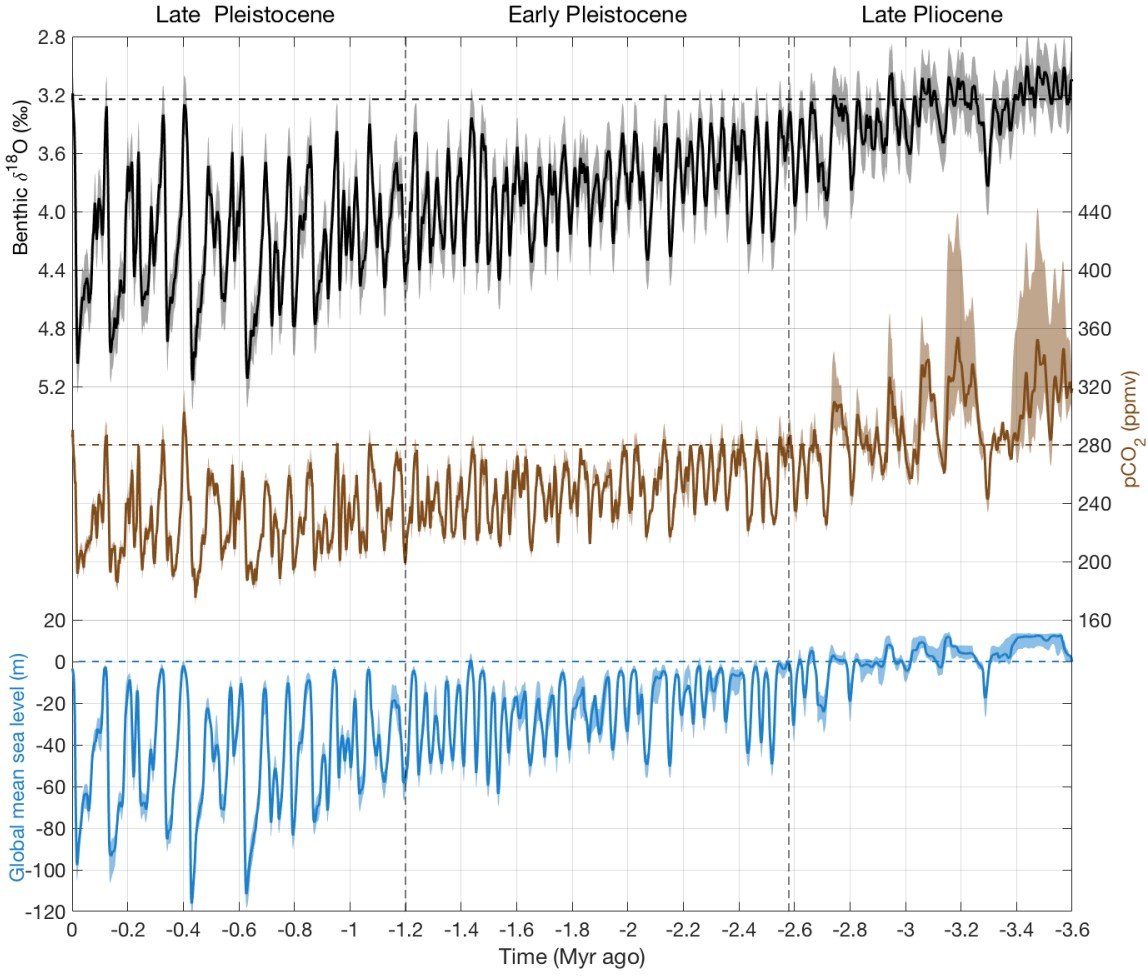

**Figure 4: Observed benthic δ¹⁸O (black; Lisiecki and Raymo, 2005) and reconstructed CO₂ (brown) and global mean sea level (blue) for the entire 3.6 Myr simulation period. The present-day values for the three variables (pre-industrial value of 280 ppmv for CO₂) are shown by horizontal dashed lines. Shaded areas indicates the uncertainty in the LR04 benthic δ¹⁸O stack, and the resulting uncertainty in the reconstructed CO₂ and sea level.**

The modelled sea-level is compared to several other reconstructions in Fig. 5. Shown are the reconstructions based on planktic δ¹⁸O in the Red Sea (Grant et al., 2014) and in the Mediterranean δ¹⁸O (Rohling et al., 2014), which cover the last 500 kyr and 5.5 Myr, respectively. The reconstruction by de Boer et al. (2014) is based on an ice-sheet model-based decomposition of the benthic δ¹⁸O signal, very similar to the work presented here. The main difference is the climate forcing applied to the ice-sheet model, which by de Boer et al. (2014) was described by a simple globally uniform temperature offset, and which was replaced

by the climate matrix approach plus inversely modelled $CO_2$ in our work. Despite this improvement in the climate forcing, which resulted in a simulated ice-sheet at LGM which agrees better with geomorphological evidence (Berends et al., 2018), the sea-level reconstructions are virtually indistinguishable. The reconstruction by Willeit et al. (2019) shown in Fig. 5 is based on a fully coupled ice-sheet – climate – carbon cycle model, forced only with insolation. However, their ice-sheet model only simulated the Northern Hemisphere; the contribution from Antarctica is assumed to be 10 % of that of the northern ice sheets. While this assumption seems to produce good results during the Pleistocene, its validity during the warmer-than-present late Pliocene is questionable. The reconstruction by Elderfield et al. (2012) was made using a Mg/Ca-based reconstruction of sea surface temperature, which was then used to disentangle the ice volume and deep-sea temperature signals in the benthic $\delta^{18}O$ record. Lastly, the reconstruction by Naish et al. (2009), adjusted by Miller et al. (2012) to match results from geological backstripping in New Zealand, suggests a relatively stable sea-level during the late Pliocene, 3.4 – 2.58 Myr ago, which agrees well with our results. Rohling et al. (2014), arguing that geological backstripping can provide information on relative changes in sea-level but not on absolute values, added a +20 m offset to the results by Miller et al. (2012), which made those results agree better with their own reconstruction from the Mediterranean Sea. However, our own results are reasonably close to those by Miller et al. (2011) over the period between 3.4 and 2.6 Myr ago without requiring such a correction. The strong variability in global mean sea-level visible in the results of Rohling et al. (2014) during this period, which suggests the repeated disappearance and reappearance of most of the East Antarctic ice sheet, is not visible in the data of Miller et al. (2012), nor in our own results.

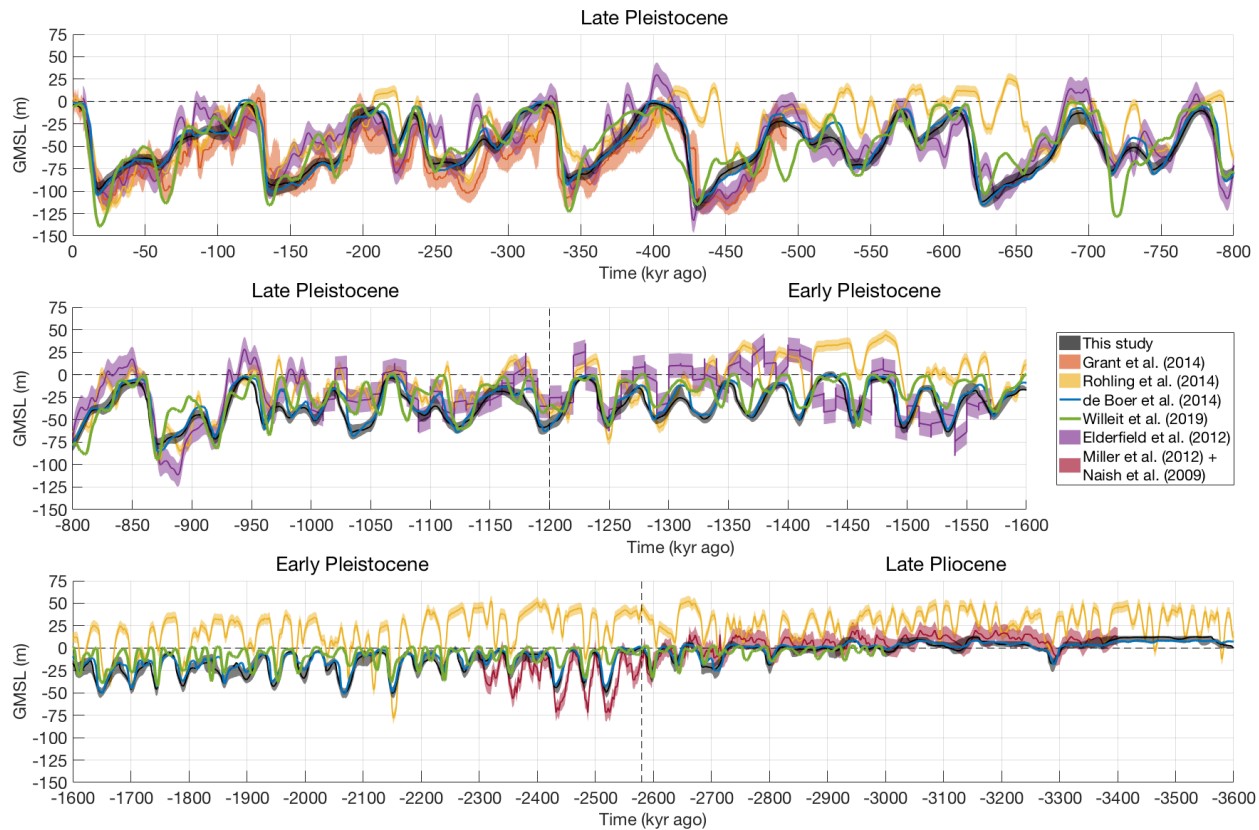

**Figure 5: Reconstructed global mean sea-level for the entire simulation period (black), compared to reconstructions based on Red Sea δ18O (Grant et al., 2014; red), Mediterranean δ18O (Rohling et al., 2014; yellow), an ice-sheet model-based inversion of the global benthic δ18O similar to this study (de Boer et al., 2014; blue), an insolation-forced, fully coupled ice-sheet – climate – carbon-cycle model (Willeit et al., 2019; green), a separation of the ice- and temperature-induced δ18O signals based on Mg/Ca ratios (Elderfield et al., 2012; purple), and a direct scaling of benthic δ18O scaled to match results from geological backstripping (Miller et al. 2012; Naish et al. 2009; dark red). The present-day value of zero is shown by a dashed line.**

The $CO_2$ reconstruction for the past 800 kyr is compared to the EPICA Dome C ice core record (Bereiter et al., 2015) in Fig. 6, as well as to several different model reconstructions and proxy-based reconstructions. The three other model-based reconstructions shown were created by decoupling the benthic δ18O signal using a 1-D ice-sheet model (van de Wal et al., 2011; Stap et al., 2017), or by using an insolation-forced, fully coupled ice-sheet – climate – carbon cycle model (Willeit et al., 2019). The geological proxies are based either on alkenones (Seki et al., 2010; Badger et al., 2013; Zhang et al., 2013) or δ11B ratios (Hönisch et al, 2009; Seki et al., 2010; Bartoli et al., 2011; Martínez-Botí et al., 2015; Stap et al., 2016; Chalk et al., 2017; Dyez et al., 2018; Sosdian et al., 2018) derived from benthic foraminifera, or based on stomata (Kürschner et al., 1996; Stults et al., 2011; Bai et al., 2015; Hu et al., 2015).

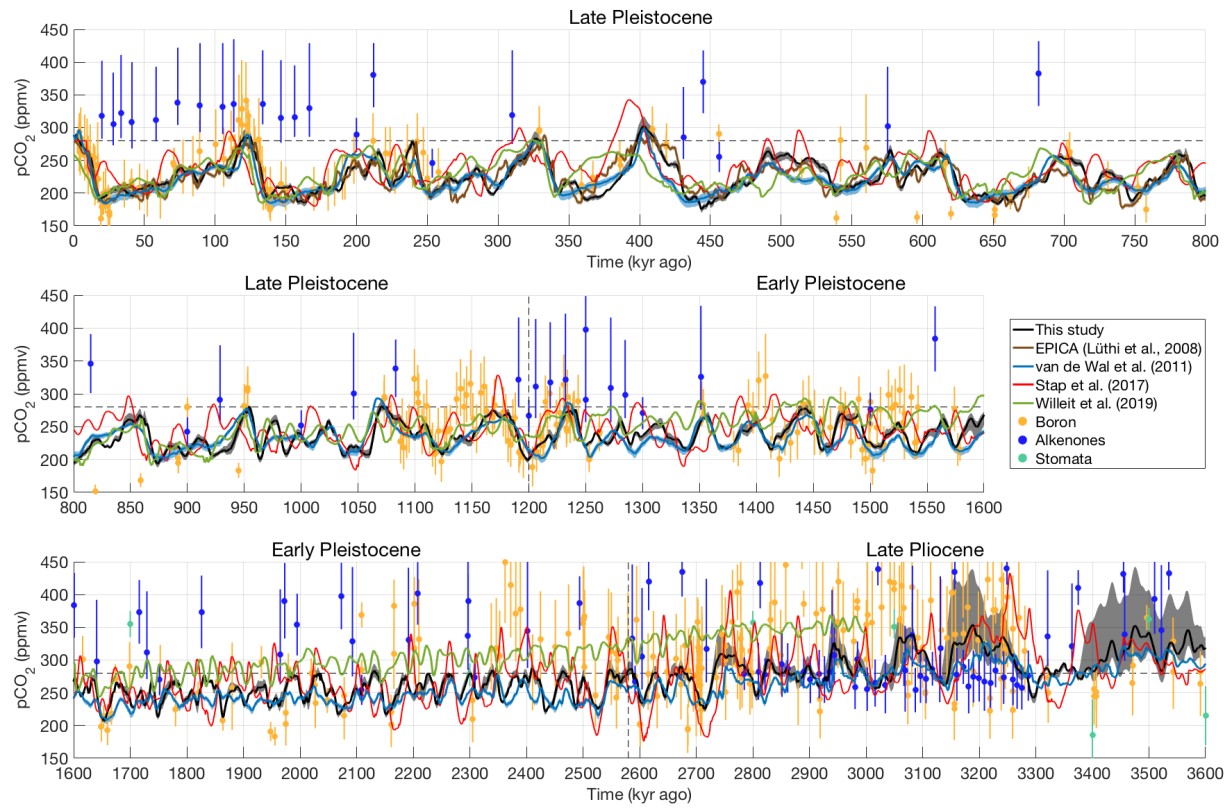

**Figure 6: Reconstructed atmospheric CO₂ for the entire simulation period, compared to the EPICA Dome C ice core record (Bereiter et al., 2015), three different reconstructions based on ice-sheet-climate models (van de Wal et al., 2011; Stap et al., 2017; Willeit et al., 2019), as well as to chemical proxies based on ¹¹B ratios (Hönisch et al., 2009; Seki et al., 2010; Bartoli et al., 2011; Martínez-Botí et al., 2015; Stap et al., 2016; Chalk et al., 2017; Dyez et al., 2018; Sosdian et al., 2018), alkenones (Seki et al., 2010; Badger et al., 2013; Zhang et al., 2013) and stomata (Kürschner et al., 1996; Stults et al., 2011; Bai et al., 2015; Hu et al., 2015). The pre-industrial value of 280 ppmv is shown by a dashed line.**

Our results broadly match those of van de Wal et al. (2011), who reconstructed CO₂ using 1-D ice models, forced with a simple spatially uniform temperature offset (assuming the strength of all feedbacks in the CO₂-temperature relation to be constant in time) calculated from benthic $\delta^{18}$O using a similar inverse forward modelling approach. Both our results and those of van de Wal et al. (2011) show values during the late Pliocene that were higher than pre-industrial, but lower than present-day (412 ppmv at the date of writing), though the values by van de Wal et al. (2011) are at the low end of the uncertainty of our own results, as can be seen in Fig. 6. Stap et al. (2017) used a 1-D ice-sheet model set-up very similar to that by van de Wal et al. (2011), but used a more elaborate energy-balance model to represent the global climate. Their results differ markedly from ours, and those of van de Wal et al. (2011). The reconstruction by Willeit et al. (2019), who used a coupled ice-sheet – climate – carbon cycle model, agrees with ours and that by van de Wal et al. (2011) in terms of the glacial-interglacial amplitude both before and after the MPT. However, the downward linear trend in pCO₂, which Willeit et al. (2019) prescribed manually to

induce the inception of northern hemisphere glaciation at 2.6 Myr ago, and the resulting high values during the Early Pleistocene, are not visible in the other reconstructions.

None of the model-based reconstructions agrees better than the others with the available proxy data. A recent study by Badger et al. (2019) showed that the alkenone proxy has only a very low sensitivity to atmospheric $CO_2$ at low to moderate $CO_2$

concentrations, as is clearly visible in the top panel of Fig. 6, where the alkenone proxy data show a nearly constant $CO_2$ concentration throughout the last glacial cycle. They state that the reliability of this proxy for values lower than ~350 ppmv (the entire Pleistocene and large parts of the Late Pliocene) is doubtful. In addition, several studies (Indermühle et al., 1999; Jordan, 2011; Porter et al., 2019) have questioned the reliability of the stomatal proxies. However, even when we disregard both of these proxies, the level of disagreement between the different boron isotope-based proxies is such that using them to

choose one model-based reconstruction over another is not possible. The only available data record which is reliable enough to allow such a comparison is the ice-core record (Bereiter et al., 2015). In Table 1, we compare the performance of the different model-based studies at reproducing this $CO_2$ record. All reconstructions have been prescribed an "optimised time lag" to find the highest possible correlation with the ice core record. Our results show the best agreement in terms of the coefficient of determination $R^2 = 0.74$, the root-mean-square-error RMSE = 13.5 ppmv (both taken over the entire 800 kyr ice-core period),

and zero time lag required to obtain these values. To put these numbers into perspective, the same numbers are listed for a simple least-squares linear fit between the LR04 benthic $\delta^{18}O$ stack and the EPICA Dome C record. This statistical "model" yields approximately the same results as our reconstruction in terms of correlation and RMSE, although the resulting post-MPT glacial-interglacial amplitude is about 23 ppmv too small, indicating that the increased glacial-interglacial variability is not properly captured.

**Table 1: Statistical comparison of the different model-based $CO_2$ reconstructions (this study; van de Wal et al., 2011; Stap et al., 2017; Willeit et al., 2019) to the EPICA Dome C ice core record (Bereiter et al., 2015), as well as the glacial-interglacial amplitude in $CO_2$ both before and after the MPT for the different reconstructions.**

|  | EPICA | Linear regression | This study | Wal2011 | Stap2017 | Willeit2019 |
|---|---|---|---|---|---|---|
| Optimised time lag (kyr) | - | - | 0.0 | 2.0 | 10.0 | 9.0 |
| Model-EPICA $R^2$ | - | 0.70 | 0.74 | 0.64 | 0.48 | 0.45 |
| Model-EPICA RMSE (ppmv) | - | 13.9 | 13.5 | 15.8 | 24.9 | 19.9 |
| Pre-MPT amplitude (ppmv) | - | 32 | 48 | 34 | 83 | 37 |
| Post-MPT amplitude (ppmv) | 91 | 68 | 87 | 79 | 104 | 72 |

Fig. 7 illustrates the difference between our modelled $CO_2$ values for the Early and Late Pleistocene. Interglacial $CO_2$ decreases from 280 – 300 ppmv at the beginning of the Early Pleistocene, to 250 – 280 ppmv just before the MPT. Glacial $CO_2$ decreases from 220 – 250 ppmv to 205 – 225 ppmv during this period, indicating a background trend of about -20 ppmv in 1.5 Myr. After the MPT, interglacial $CO_2$ varies between 250 – 320 ppmv, and glacial values decrease to 170 – 210 ppmv, with neither

range showing a clear trend. The reduced glacial-interglacial $CO_2$ difference of 48 ppmv (averaged over the entire period) that we find in our results agrees with the value of 43 ppmv observed in the boron isotope data by Chalk et al. (2017). The values for all four model-based reconstructions, both before and after the MPT, are shown in Table 1. After the MPT, we find a glacial-interglacial difference of 87 ppmv, which agrees very well with the value of 91 ppmv over the past 800 kyr from the

EPICA Dome C ice core record (Bereiter et al., 2015).

The discussion about the nature of the Late Pleistocene glacial cycles is still ongoing. Different studies have proposed that they represent either a strongly amplified response to a weak 100 kyr forcing (Ganopolski and Calov, 2011), a response to the same 40 kyr forcing as in the Early Pleistocene, with an additional threshold mechanism that causes some insolation maxima to be "skipped", leading to 80/120 kyr cyclicity (Huybers, 2011; Tzedakis et al., 2017), or even as the result of some stochastic

process with a collapse threshold, containing no true periodicity (Wunsch, 2003). Since the timing of glaciations and deglaciations in our model is forced by the LR04 benthic $\delta^{18}O$ stack, and thus its depth-age model, no meaningful conclusions regarding this discussion can be drawn from our results.

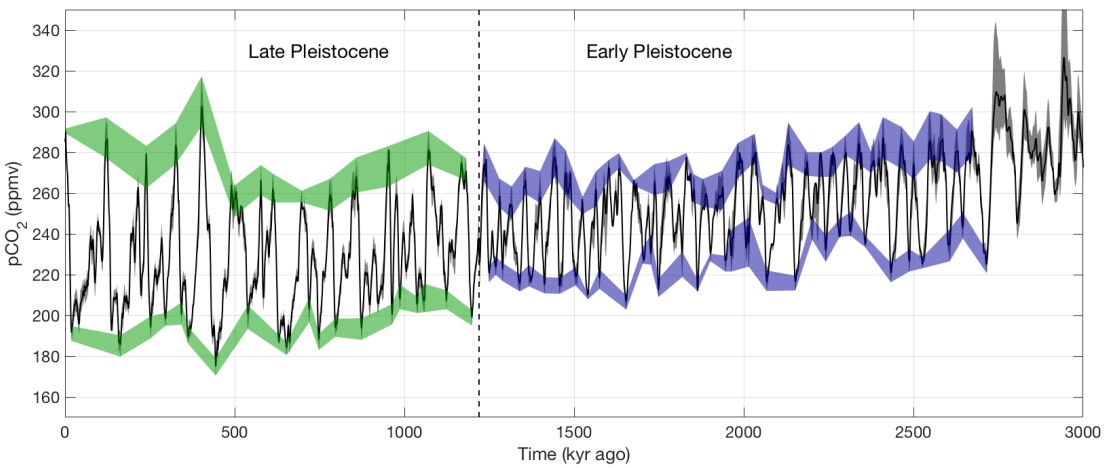

**Figure 7: Reconstructed atmospheric $CO_2$ during the Pleistocene. The coloured regions indicate the glacial and interglacial ranges**
**for the Early (2.6 – 1.2 Myr ago; blue) and Late (1.2 – 0.011 Myr ago; green) Pleistocene.**

The non-linear, time-dependent relation between $CO_2$ and sea level is visualised in Fig. 8, both for the different proxy data (panel A) and the different model reconstructions (panels B-F) using scatter plots. This visualisation contains several useful features; the general shape of the curve shows the sensitivity of the world's ice sheets to changes in climate, while the spread of the data points around the curve indicates the time lag between the two. An instantaneous response would result in zero

spread, while a long lag would yield a wide spread. This time lag is caused by the slow response of ice sheets to climate change, the intrinsic hysteresis of global ice sheet volume under a changing climate, and other slow-responding physical processes such as englacial thermodynamics, glacial-isostatic adjustment, and oceanic carbon cycling.

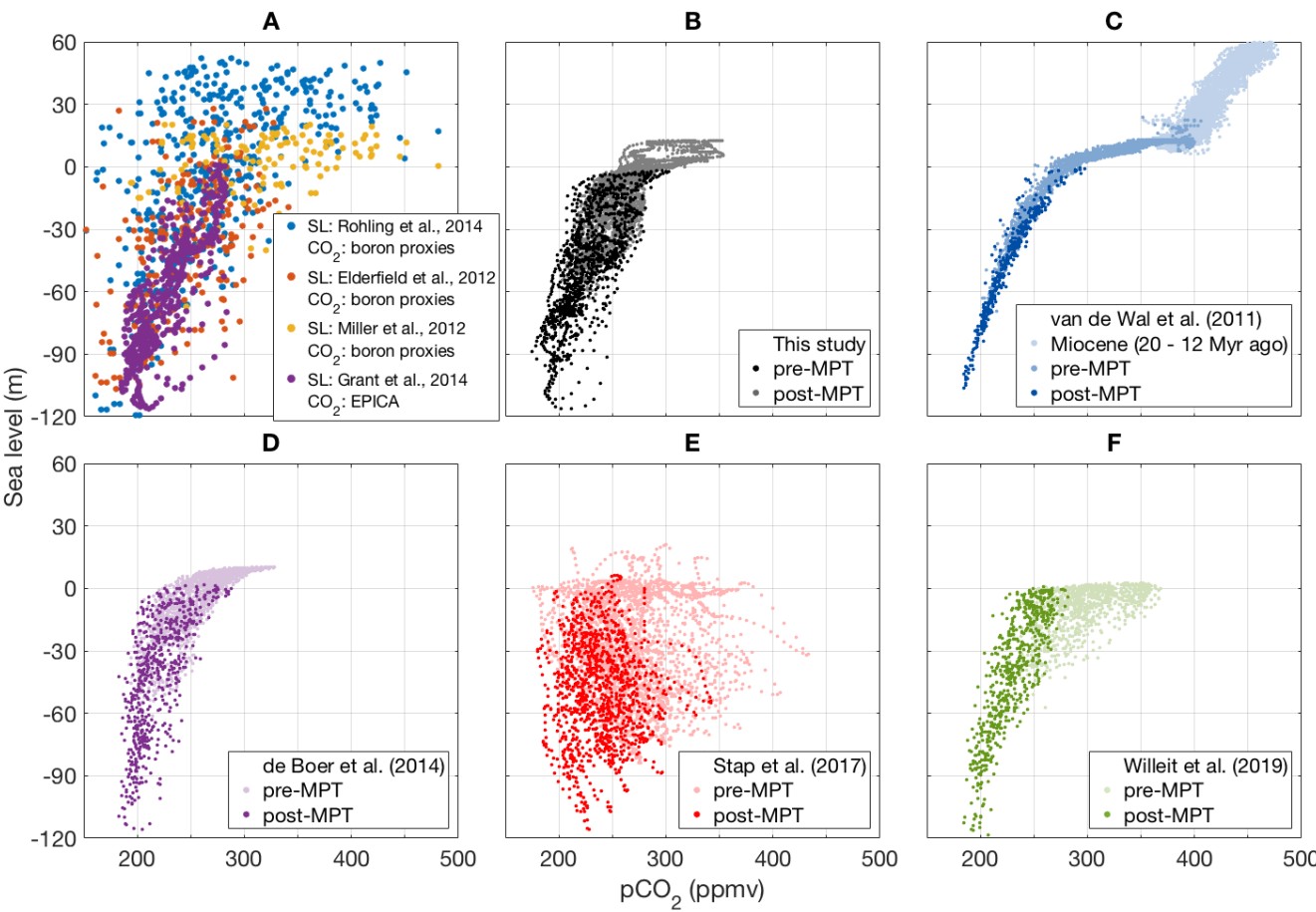

**Figure 8: A: Sea level vs atmospheric CO₂ from proxy data. Blue: sea-level data from Rohling et al. (2014; last 5 Myr) vs boron isotope CO₂ data. Red: sea-level data from Elderfield et al. (2012; last 1.5 Myr) vs boron isotope CO₂ data. Gold: sea-level data from Miller et al. (2012) and Naish et al. (2009; 3.4 – 2.3 Myr ago) vs boron isotope CO₂ data. purple: sea-level data from Grant et al. (2014; last 500 kyr), vs the EPICA Dome C ice core record (Bereiter et al., 2015). B-F: reconstructed sea level and atmospheric CO₂ from different modelling studies (B: this study; C: van de Wal et al., 2011; D: de Boer et al., 2014; E: Stap et al., 2017; F: Willeit et al., 2019), separated into pre- and post-MPT values. de Boer et al. (2014; panel D) did not explicitly model CO₂, calculating only a global mean temperature change. We converted this to CO₂, using the logarithmic relation from van de Wal et al. (2011).**

The combination of ice core CO₂ data (Bereiter et al., 2015) and Red Sea sea-level data (Grant et al., 2014), regarded as the two most reliable proxy reconstructions for the colder-than-present worlds of the Late Pleistocene and shown by the purple data points in Panel A, displays a pattern that is very similar to the model results by de Boer et al. (2014), and by this study. The reconstruction by van de Wal et al. (2011) shows a narrower range of sea levels for each CO₂ value, indicating that their ice sheets respond faster to changes in climate. The reconstruction by Willeit et al. (2019) show a wider spread in the 250 – 350 ppmv range, implying the existence of medium-sized ice sheets (~30 m SLE) even when CO₂ is above 280 ppmv.. The reconstruction by Stap et al. (2017) shows a vast spread, indicating that their ice sheets respond very slowly. The cause of this is not known (*personal communication with authors*). While both the observational data in Fig. 8A, and the results by Willeit et al. (2019) show the existence of medium-sized ice sheets under CO₂ concentrations as high as 300 ppmv, the results by van

de Wal et al. (2011), de Boer et al. (2014), and this study all show a "threshold" for glaciation and sea-level drop around 250 ppmv. This indicates that the effects of orbital changes on the global climate, and by extension on the surface mass balance and the timing of glacial inceptions, are underestimated in these models.

The much weaker response of sea level to changes in $CO_2$ for concentrations higher than 280 ppmv, which is displayed by all models except Stap et al. (2017), seems to be confirmed by the sea-level data by Miller et al. (2012) and the boron isotope $CO_2$ data. The sea-level data by Rohling et al. (2014) show much more variation, especially during the early Pleistocene. The results by van de Wal et al. (2011) show a very clear threshold around 375 ppmv $CO_2$, beyond which sea level rises rapidly with increasing $CO_2$. As indicated by the colours, they only find these warm greenhouse worlds during the first half of the Miocene (20 – 12 Myr ago), which explains why none of the other models, nor the proxy data, display this behaviour. Although some of the boron isotope proxies suggest $CO_2$ concentrations higher than this 375 ppmv threshold value, the sea level data do not show the associated strong rise.

## 5 Conclusions and discussion

We have presented a new, time-continuous, self-consistent reconstruction of atmospheric $CO_2$, ice sheet evolution and global mean sea-level of the last 3.6 Myr, based on the benthic $\delta^{18}O$ glacial proxy. This reconstruction was created by using a hybrid ice-sheet – climate model (Berends et al., 2018) to decouple the contributions from ice volume and deep-water temperature to the benthic $\delta^{18}O$ record (Lisiecki and Raymo, 2005). Our sea-level reconstruction agrees well with similar model-based results (de Boer et al., 2014), the reconstruction based on Red Sea $\delta^{18}O$ by Grant et al. (2014) and the combined results of benthic $\delta^{18}O$ (Naish et al., 2009) and geological backstripping (Miller et al., 2012). Our results agree less well with the reconstruction based on Mediterranean Sea $\delta^{18}O$ by Rohling et al. (2014), which goes back further in time but has a lower signal-to-noise ratio, and with the reconstruction based on benthic $\delta^{18}O$ and Mg/Ca ratios by Elderfield et al. (2012). Although previous studies (van de Wal et al., 2011; de Boer et al., 2014; Stap et al., 2017) used more simple, parameterized ice-sheet and climate models, the resulting sea-level reconstructions are very similar.

Our $CO_2$ reconstruction agrees well with the EPICA Dome C ice core record (Bereiter et al., 2015), showing the strongest correlation of all four model-based reconstructions. Our results do not agree well with different chemical proxy-based reconstructions based on foraminiferal alkenones (Seki et al., 2010; Badger et al., 2013; Zhang et al., 2013), foraminiferal $\delta^{11}B$ ratios (Seki et al., 2010; Martínez-Botí et al., 2015; Stap et al., 2016; Chalk et al., 2017; Dyez et al., 2018; Sosdian et al., 2018) or fossil plant stomata (Kürschner et al., 1996; Stults et al., 2011; Bai et al., 201; Hu et al., 2015). However, the strong spread between those different proxies and the large uncertainty of each individual proxy record do not allow to conclude whether the difference between those proxies, and our model reconstructed values is significant.

Between 2.8 and 1.2 Myr ago, during the early Pleistocene, we simulate 40 kyr glacial cycles with glacial-interglacial sea-level changes of 25 – 50 m. During this period, $CO_2$ varies between 270 – 280 ppmv during interglacials and 210 – 240 ppmv

during glacial maxima, with both values decreasing by about 20 ppmv over the 1.4 Myr course of the Early Pleistocene. After the Mid-Pleistocene Transition (MPT), when the glacial cycles change from 40 kyr to 80/120 kyr cyclicity, their sea-level amplitude increases to $70 - 120$ m, with $CO_2$ varying between $250 - 320$ ppmv during interglacials and $170 - 210$ ppmv during glacial maxima. This implies a glacial-interglacial contrast of about 45 ppmv pre MPT and 85 ppmv post MPT. The two most

reliable proxy records, the sea-level reconstruction by Grant et al. (2014) and the EPICA Dome C ice core record (Bereiter et al., 2015), display a relation between sea-level and $CO_2$ for colder-than-present worlds that matches our model results. For warmer-than-present worlds, the large spread and uncertainty in available proxy data for both sea-level and $CO_2$, as well as the much larger uncertainty in our model results during these warm periods, prevent us from drawing conclusions on the reliability of our methodology in this regard. We believe that the inverse modelling method of using benthic $\delta^{18}O$, combined

with coupled or hybrid ice-sheet – climate models, as a proxy for past $CO_2$ and sea-level, can add valuable insights into the evolution of the Earth system during these warm episodes.

During the warm late Pliocene, we find a large variability in $CO_2$, with a difference of about 100 ppmv between the cold glacial excursion 3.3 Myr ago, and the warm peak around 3.2 Myr ago. The boron isotope-based data by Martínez-Botí et al. (2015),

which has the highest temporal resolution and longest temporal range of all proxy records, shows a variability of about 150 ppmv during this period (albeit with an uncertainty of about 100 ppmv in either direction). While the large discrepancies between different proxy records are too large to draw any definitive conclusions, their findings do seem to support strong $CO_2$ variability during these warm periods. Our results for this period also show a much larger uncertainty range than during the Pleistocene. We suspect that the relative sparsity of our current matrix in this warm regime biases the model towards large ice

sheets in warm worlds. The high variability in our reconstructed $CO_2$ during the late Pliocene can then be explained as a consequence of this bias, and the requirement posed by the inverse modelling approach that the prescribed $\delta^{18}O$ record is reproduced. If a high (warm, little ice) $\delta^{18}O$ is prescribed, but little or no ice-sheet retreat occurs in the model, then this must be compensated for by an additional increase in deep-water temperature, which requires large changes in surface climate, and therefore in $CO_2$. By adding additional GCM snapshots for worlds with $> 400$ ppmv $CO_2$ and smaller-than-present ice sheets

(particularly a reduced East Antarctic ice sheet), modelled ice-sheet retreat in warm climates might be increased, reducing simulated $CO_2$ variability during these periods. While this means that the modelled sea-level rise during this period will be larger, our current results are at the low end of the uncertainties of other reconstructions, implying that a higher modelled value will not immediately lead to a mismatch.

The parameterised relation between the global climate, deep-sea temperature and the resulting contribution to the $\delta^{18}O$ signal could be improved upon. In this study, we used the parameterisation developed by Bintanja and van de Wal (2008), which is based on the fact that the LR04 stack consists predominantly of Atlantic records, and on the assumption that Atlantic deep-water temperature is strongly related to annual mean surface temperature in the North Atlantic, where deep water is formed by

downwelling. This constitutes a strong simplification of the complex relation between atmospheric and oceanic temperatures, and one that is no longer necessary in our model, where the globally uniform temperature offset used by Bintanja and van de Wal (2008) has been replaced with the GCM snapshots used in our matrix method. Based on output from available ocean-enabled GCM simulations of the LGM, a more elaborate parameterisation could be constructed, perhaps using surface

temperatures over specific downwelling regions. Instead of using a global benthic $\delta^{18}O$ stack, it might even possible to separate the records from different ocean basins.

Although the margins of error we report for our $CO_2$ and global mean sea-level reconstructions are relatively small, these describe only the uncertainty arising from the uncertainty in the prescribed $\delta^{18}O$ forcing. The true uncertainty in our

reconstructions depends on many other factors, including the many parameterisations included in our model (such as the matrix method, and the relation between atmospheric and oceanic temperatures, and benthic $\delta^{18}O$), and is definitely large than the margins of error reported here. However, the uncertainties arising from these factors an unfortunately not be evaluated at this stage.

Our results should not be interpreted as a realistic reconstruction of what the world looked like in terms of global climate, ice sheet geometry, sea level and $CO_2$, during these periods of geological history. Rather, we believe they should be viewed as scenarios, which can help to interpret an expected new ice core record. For example, one hypothesised mechanism behind the MPT is regolith erosion, which led to a change in basal conditions from mostly sliding ice to mostly non-sliding ice over North America and Eurasia during the MPT (Clark and Pollard, 1998; Willeit et al., 2019. Basal conditions in our model are constant

over time, so that the MPT is entirely attributed to changes in $CO_2$. A simulation identical to ours, but including a change in basal conditions, would result in a different pre-MPT $CO_2$ history. Comparing the new ice core record to these two different scenarios could help determine if regolith erosion was a crucial factor in the transition. Other proposed physical mechanisms, such as regolith erosion leading to a different relation between ice sheet geometry and glaciogenic dust, changes in ocean circulation leading to a different dynamic equilibrium between atmospheric and oceanic carbon, or a slow background decrease

in $CO_2$ caused by weathering or reduced volcanism, could be explored in a similar manner. We believe our modelling approach is very suited for this kind of exploratory analysis.

*Data availability.* The reconstructed global mean sea-level and atmospheric $CO_2$ concentration over the past 3.6 Myr are available online at doi.org/10.5281/zenodo.3793592

**Appendix A**

Here, we briefly summarise the equations governing the matrix method. A more thorough explanation of this approach can be found in the original publication by Berends et al. (2018).

Each GCM snapshot contains data fields describing global monthly mean 2-m air temperature, global monthly mean precipitation, and surface elevation. The interpolation routine for the temperature field uses modelled $CO_2$ and modelled "absorbed insolation" as weighting parameters to interpolate between snapshots. This approach captures the two most important physical processes through which a change in ice-sheet geometry affects the local and global temperature; the ice-albedo feedback, and the altitude-temperature feedback. Since the absorbed insolation changes not only through changes in albedo, but also through changes in incoming insolation, the effects of orbital forcing are also (indirectly) accounted for. The weighting factor $w_{CO2}$ is calculated as:

$$w_{CO2} = \frac{pCO_2 - pCO_{2,LGM}}{pCO_{2,PI} - pCO_{2,LGM}}, \tag{A1}$$

with $pCO_{2,PI} = 280$ ppmv and $pCO_{2,LGM} = 190$ ppmv. Multiplying the modelled surface albedo $\alpha$ with the prescribed insolation at the top of the atmosphere $Q_{TOA}$ yields the absorbed insolation $I_{abs}$:

$$I_{abs}(x, y) = (1 - \alpha(x, y)) \cdot Q_{TOA}(x, y). \tag{A2}$$

This value is scaled between the PI and LGM reference fields to obtain the weighting parameter $w_{ins}$:

$$w_{ins}(x, y) = \frac{I_{abs,mod}(x, y) - I_{abs,LGM}(x, y)}{I_{abs,PI}(x, y) - I_{abs,LGM}(x, y)}. \tag{A3}$$

This weighting field is partly smoothed with a 200 km Gaussian filter $F$, to account for both local and regional effects:

$$w_{ice,T}(x, y) = \frac{1}{7} w_{ins}(x, y) + \frac{3}{7} F(w_{ins}(x, y)) + \frac{3}{7} \overline{w_{ins}}, \tag{A4}$$

This is combined with the scalar $pCO_2$ weight $w_{CO2}$ to yield the final temperature weighting parameter $w_T$:

$$w_T(x, y) = \frac{w_{CO2} + w_{ice,T}(x, y)}{2}. \tag{A5}$$

This weighting field is used to interpolate both the temperature and orography fields of the GCM snapshots:

$$T_{ref}(x,y) = w_T \cdot T_{PI}(x,y) + (1 - w_T) \cdot T_{LGM}(x,y), \tag{A6}$$

$$h_{ref}(x,y) = w_T \cdot h_{PI}(x,y) + (1 - w_T) \cdot h_{LGM}(x,y). \tag{A7}$$

Orography is interpolated as well, so that it can be used to downscale the temperature field from the low-resolution GCM grid to the higher-resolution ice-model grid, using a simple lapse-rate approach:

$$T(x,y) = T_{ref}(x,y) + \lambda_{LGM}(x,y)\left(h(x,y) - h_{ref}(x,y)\right), \tag{A8}$$

The spatially variable lapse rate $\lambda_{LGM}(x,y)$ is derived from the temperature field of the GCM snapshot.

The interpolation for precipitation is based on ice thickness rather than absorbed albedo. This reflects the fact that precipitation over the ice sheet is strongly affected by altitude:

$$w_{ice,P}(x,y) = \frac{Hi_{mod}(x,y) - Hi_{PI}(x,y)}{Hi_{LGM}(x,y) - Hi_{PI}(x,y)} \cdot \frac{V_{mod} - V_{PI}}{V_{LGM} - V_{PI}}, \tag{A9}$$

$$w_P(x,y) = \frac{w_{CO2} + w_{ice,P}(x,y)}{2}. \tag{A10}$$

$$P_{ref}(x,y) = e^{\left(w_P \cdot log\left(P_{PI}(x,y)\right) + (1 - w_P) \cdot log\left(P_{LGM}(x,y)\right)\right)}, \tag{A11}$$

Finally, the precipitation field is downscaled from the low-resolution GCM grid to the higher-resolution ice-model grid, using the precipitation model by Roe and Lindzen (2001) and Roe (2002), to correct for the steeper slopes in the ice model:

$$P(x,y) = P_{ref}(x,y)\frac{P_{Roe}(x,y)}{P_{Roe_{ref}}(x,y)}. \tag{A12}$$

*Author contributions.* CJB, BdB, and RSWvdW designed the study. CJB created the model set-up and carried out the simulations, with support from RSWvdW. CJB drafted the paper, and all authors contributed to the final version.

*Competing interests.* The authors declare that they have no conflict of interest.

*Acknowledgements.* The Ministry of Education, Culture and Science (OCW), in the Netherlands, provided financial support for this study via the program of the Netherlands Earth System Science Centre (NESSC). B. de Boer was funded by NWO Earth and Life Sciences (ALW), project 863.15.019. This work was sponsored by NWO Exact and Natural Sciences for the use of supercomputer facilities. Model runs were performed on the LISA Computer Cluster, we would like to acknowledge

SurfSARA Computing and Networking Services for their support. Special thanks go to B. Hönisch for advice on paleo-$CO_2$ data usage.

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
