# Peer review of "Reconstructing the Evolution of Ice Sheets, Sea Level and Atmospheric CO2 During the Past 3.6 Million Years"

_Climate of the Past, 2020_

## Referee Comment (RC1) · Anonymous Referee #1 · 25 May 2020

Overall assessment

The paper by Berends et al. represents an extension of a previous study by the same authors published in CP with the aim to obtain a consistent history of CO2, ice sheet size/sea level and deep ocean temperature over the last 3.6 million years using an inverse modeling approach. As such the paper is well suited for publication in CP and the results have great potential to improve our understanding of Earth System Sensitivity and ice sheet development over several time intervals critical for the glaciation history: the Pliocene/Pleistocene Transition (PPT), the Mid Pleistocene Transition (MPT) and the Mid Brunhes Event (MBE). The paper is also well written and supported by

sufficient figures and tables. Having said that, I feel the paper in its current version does not make full use of its potential. Moreover, I have some methodological questions/suggestions for improvements, which should be addressed before acceptance. Accordingly, I recommend the paper for Major Revisions. Although these revisions may require a bit more time, I regard them as relatively straightforward.

General Comments:

Regarding my first point that the paper could become of greater impact, I raise the following questions:

a) The three transitions mentioned above represent enigma in our understanding of the glaciation history. In particular, the shift in glaciation cyclicity from the obliquity driven cycles prior to the MPT and the 80-100 kyr cycles thereafter gave rise to the so called regolith hypotheses (Clark et al., WSR 2006). This hypothesis invokes a change in the Laurentide ice sheet bed conditions that changes the ice sheet flow, hence the ice sheet cross section. Another study by Tzedakis et al. (Nature 2017) invokes a change in an energy threshold for deglaciation over the MPT caused by summer insolation. The study by Berends et al. - if I understand it correctly - does not need either of the two to get the transitions and the change in cyclicity right. The entire record (including changes in amplitude and cyclicity) is entirely controlled by CO2. In particular, it has a constant relationship between d18O_benthic and CO2 over time and does not change the flow conditions at the bed. I highly recommend that the authors elaborate on this extensively in the Discussion and the Conclusions and discuss what this potentially implies for our understanding of the glaciation history. Related to this, the authors set out in the (very nice!) introduction that these experiments will help us to understand and quantify Earth System Sensitivity. I regard it a missed chance that they do not pick up on this issue in the Conclusions as they seem to have all data at hand to contribute to this discussion. Given that the paper in its current form is relatively short, there is enough space to elaborate on this.

b) I am puzzled by the way the deep ocean temperature, which influences the d18O_benthic signal, is calculated. In the manuscript and in Berends et al. (2019) the authors say that they used the global average of the surface temperature anomaly. Either there is some detail missing here (some scaling) or this seems to be at odds with the measured deep ocean temperature today. Here it is important to take into account that deep ocean temperatures have a strong bias towards the sea surface temperatures at deep water formation sites, which are located in high latitudes. There is also eddy diffusive transport of heat in lower latitudes, but the low deep water temperatures clearly point to a dominating role of deep water formation. In fact, the resulting deep ocean temperature, which is caused by the balance between advective transport of cold water from deep water formation and the diffusive entrainment of heat downwards, is also dependent on the strengths of the Atlantic Meridional Overturning Circulation (Galbraith et al., GRL 2016), which is also not included in the approach by Berends et al. The value of the deep ocean temperature used in the study by Berends is not mentioned in the paper. It is likely too warm, but this is also likely compensated by the CO2 sensitivity (120 ppm/permille) with which their approach is optimized. Even if there is some scaling involved that is not described in the manuscript, using the global average surface temperature appears to be an oversimplification. Accordingly, I think it is important to discuss this issue and use some alternative sensitivity runs, where the deep ocean temperature is parameterized by high latitude temperatures and the model CO2 sensitivity is recalibrated to show that the final result is not sensitive on the choice of the deep water temperature template.

Specific comments:

abstract line 2: "...over geological time scales... "

abstract line 3: "...past CO2 concentrations, thus its radiative forcing, only... "

abstract line 16-17: please do not use unexplained abbreviations such as KM5c and M2 in the abstract. Please also indicate such stages after you introduced them in the

main text in the figures.

page 2 line 4: "... atmosphere from the time..."

page 2 line 6 and throughout the manuscript: Here you cite Bereiter et al. (2015) for the $CO_2$ record, but later you cite Lüthi et al., 2008. Note that in Bereiter et al., a correction of the EPICA Dome C $CO_2$ values by Lüthi et al. was introduced for ice older than about 600 kyr. You should use the corrected Bereiter et al. data throughout the manuscript.

page 2 line 9: "... have measured d11B..."

page 4 line 14: "... can help interpreting..."

page 6 line 16: Here the d18O/CO2 scaling parameter is introduced. While it has been mathematically introduced in equation 1, it would be helpful to discuss the meaning of this parameter in more detail and also discuss what it implies if this parameter is assumed to be constant over time.

page 8 line 14: it is not entirely clear to me what you mean by "combining the GCM snapshots according to position of the ice-sheet model in the climate matrix". I am sure there is an easier way to explain this.

page 10 line 6: "... which cover the last 500 kyr and 5.5 Myr, respectively."

page 11, line 2-4: here you say that the model study by Willeit does not include the Antarctic ice sheet, but the results look quite similar. What does this imply?

page 11 line 10: "... close to those..."

page 12 line 9: "d11B"

page 12 caption Figure 6: The references are assigned wrongly in the caption. Hönisch et al. (2009) and Bartoli et al. (2015) use d11B, not alkenones. It is correctly referenced in the main text.

page 13 line 10: You write "which they prescribed". Who is "they" in this case, please provide the reference.

page 13 line 19: "... different boron isotope based records is such..."

page 15 Figure 8: It would be helpful if the data younger than 800 kyr and those older than 800 kyr could be discerned in the model runs. Use different symbols or colors.

page 15 line 12: Here you say that the reconstructions by van de Wal and Willeit have a smaller spread. Do you mean in CO2 or sea level? In particular, for the reconstruction by Willeit I do not see a significantly smaller spread except that the sea level is capped.

page 16 line 21: "d11B"

page 16 line 22-24: Here you say that the data/model comparison in terms of CO2 is not conclusive, but before you showed that the highest resolution d11B data by Chalk et al., show perfect agreement with your reconstructions. I think you undersell the d11B values. Clearly each individual d11B based CO2 value has an analytical uncertainty on the order of 20 ppm, but measured in high enough resolution/replication, these data are quite useful to validate your model results.

page 16 line 32: "... for a colder-than-present..."

page 17 line 4: The last sentence is weak and does not give credit to the work performed in this study. The authors should elaborate much more on this, as outlined in my general comments above.

---

## Referee Comment (RC2) · Andrey Ganopolski (Referee) · 19 Jun 2020

The manuscript by Berends et al. presents a new step in the application of inverse modelling approach in conjunction with forward modelling to study past climate variability, the approach which the Utrecht group explores already over a long time. I believe that this is a very promising approach which will allow us to learn more about past climate dynamics and internal consistency of different paleoclimate reconstructions. Unfortunately, I have a number of problems with this manuscript which require clarifications and critical discussions. I also believe that it is absolutely crucial to properly estimate the real uncertainties of the proposed method.

General comments

1. Method description. One of the problems for the readers of this manuscript is that the method used in this study has been developed over a long period time and its comprehensive description are scattered among a number of previous publications. Even although I was familiar with some of them, it took me a lot of time to get a more or less clear understanding of what authors are doing. Of course, one cannot expect such efforts from a typical reader. However, without a proper understanding of the method, the results presented in the manuscript are not very useful. This is why, I would suggest to make a more detailed description (including the key equations) in the appendix or supplementary information. In particular, I am curious how the effect of orbital forcing has been accounted for by the "matrix method".

2. The model validation is based on the comparison of reconstructed $CO_2$ over the past 800 kyr with the ice core data. The authors compare the results of their current study with several others and conclude that they are the best. However, it is obvious that comparison results of inverse modelling with forward modelling presented in Willeit et al (2019) is the same as comparison of apples with cucumbers. The inverse model is forced by benthic d18O which is already highly correlated with $CO_2$ (correlation coefficient is 0.86). The authors should make this point very clear. The only surprising thing in this table is the extremely poor performance of Stap et al. (2017). Unfortunately, the authors themselves admit on page 15 that they cannot explain this fact.

In fact, it is much more instructive to compare the result of a rather complex inverse modelling approach used by the authors to a simple linear regression

$CO_2=175+50.2 (5.2-s)$,

where "s" is 5000-years averaged d18O from LR04 stack. Surprisingly (or maybe not) this simple "model" outperforms Berends et al. Indeed, it has $R^2=0.71$ (versus 0.68 in Berends et al.) and rms=13.8 ppm (vs. 15.3) for "simulated" $CO_2$ concentration over the last 800 kyr. After such a comparison, the numbers in Table 1 do not look

very impressive. For the rest of Quaternary, results of Berends et al. also do not differ much from this simple regression model. After all, it is rather expectable (and have been demonstrated by Willeit et al., 2019) that CO2 also followed ice volume variations during 41-kyr world but with a smaller amplitude. The real question is what was CO2 concentration at the end of Pliocene. And here I see a real problem with the results presented in Berends et al. Indeed, if during the entire Pleistocene, CO2, ice volume and d18O variations were essentially identical, during the late Pliocene CO2 get really wild. Figure 4 shows several CO2 oscillations with the amplitude above 100 ppm. Of course, this is not 200+ ppm as in Stap et al (2017) but still a lot. As the scientist who has been heavily involved in explaining glacial-interglacial CO2 variability, I must confess that it is extremely difficult to explain 80 ppm change in CO2 concentration even for the full glacial cycles of the late Quaternary. What could cause even larger Pliocene variations in CO2 without any obvious external forcing, the authors do not explain. This is why I strongly suspect that the reason for such weird behaviour of CO2 before Pliocene-Pleistocene transition is that the inverse modelling of CO2 concentration based on benthic d18O beyond Quaternary represents an ill-posed problem.

3. The authors wrote on page 9 that "uncertainties are conservative in this study". What the authors mean under "conservative" is not clear to me. To me, the estimate of uncertainties in this study is overoptimistic at best. Even if the maximum error in benthic d18O is indeed only 0.1 promile, the methodology has a number of other uncertainties related both to forward model and to conversion between climate characteristics (ice volume, temperature) and d18O. For the large glacial cycles of Quaternary even a larger uncertainty still does not prevent a reasonable estimate of CO2 but the situation is very different prior to 2.7 Ma. Before Quaternary, the model "assumes" very little variability in global ice volume and thus most of d18O variability has to be attributed to CO2 change and this is precisely what the model does. However, in this case, even uncertainty of +-0.1 promile already constitutes a serious problem. Indeed, 0.2 promile correspond to about 1C change in the deep-water temperature which in turn corresponds to 1.5C in global air temperature. The later number corresponds to change of

CO2 (assuming climate sensitivity =3C) from 280 to 400 ppm. Thus, even with a very optimistic estimate of the method uncertainty, for pre-Quaternary climates this method cannot distinguish between a possibility that CO2 was as low as the preindustrial one or that it was as high as the current one. Obviously, such "reconstruction" is not very helpful. 4. "80/120 kyr cycles". Although this is not very essential for the manuscript under consideration, but the authors used the expression "80/120 kyr cycles" (actually it should be 82/123) several times in this and previous papers which provokes me to make the following comment: The durability of "two or three obliquity cycles" myths is amazing since it is not supported by real data! Glacial cycles of the late Quaternary have average periodicity close to 100 kyr which explains strong 100 kyr peak in the frequency spectra of ice volume. It is true that the durations of individual glacial cycles deviate significantly from 100 kyr but they also do not cluster around 80 and 120 kyr (see for example Table 1 in Konijnendijk et al., 2015). In fact, durations of individual glacial cycles are relatively uniformly distributed between 80 and 120 kyr with half of the cycles been closer to 100 kyr than to 80 or 120 kyr.

Specific comments

P.3, L.8 "proxies for global mean temperature"? Greenland and Antarctic records present proxies only for local temperatures which differ significantly from the global one

P3., L.10. "In that case ocean water temperature can be resolved as closure term from the benthic signal" This is not clear

P. 4, L.9 The definition of "entire climate system (atmosphere, ocean, cryosphere, carbon cycle, etc.)" is not consistent with contemporary terminology. Such system is named Earth system and Earth system models describe not only "physical processes" (L. 10).

p. 4, L. 21 "the known relations between atmospheric CO2, global temperature and climate, and ice-sheet evolution". Why authors think that these relations are "known".

Even the relation between CO2 and global temperature is still not well-known.

p. 5. I am not sure I understand why the authors put "data" and "model" in quotes.

P.7 L. 11 "The reconstruction by Laskar et al. (2004) is used to prescribe time- and latitude-dependent insolation". Insolation is not reconstructed by computed using physical laws. This is why orbital forcing can be calculated for the past and future with the same (very high) accuracy.

P. 11, L. 2. "so any possible contribution from Antarctica to changes in sea-level ... is not accounted for in their reconstruction". This is an incorrect statement. It is written in Willeit at al. (page 6) "Sea level is computed from the volume of modeled NH ice sheets assuming an additional 10% contribution from Antarctica".

P. 15, L. 15. "... show a CO2 "threshold" for glaciation and sea-level drop around 250 ppmv". Our studies (e.g Ganopolski et al., 2016) do not support the existence of a single CO2 threshold for glaciations. To the contrary, we found that glacial inception is determined by a combination of insolation and logarithm of CO2 concentration.

Fig. 4. It is not explained what shading shows in this figure.

The reference Stap et al. (2017) is not in the reference list.

---

## Author Comment (AC1) · 7 Jul 2020

Rebuttal to the review by Anonymous Referee 1

We thank the reviewer for their comments on the manuscript and would hereby like to address the concerns they raised.
Comments in italics, below our rebuttal. Page and line numbers refer to the revised manuscript.

*The study by Berends et al. - if I understand it correctly - does not need either*

[Figure]

*of the two to get the transitions and the change in cyclicity right. The entire record (including changes in amplitude and cyclicity) is entirely controlled by CO2. In particular, it has a constant relationship between d18O_benthic and CO2 over time and does not change the flow conditions at the bed. I highly recommend that the authors elaborate on this extensively in the Discussion and the Conclusions and discuss what this potentially implies for our understanding of the glaciation history.*

We use the "inverse modelling method" that does indeed use the benthic d18O record as forcing to calculate a CO2 history, but this is not based on a constant relationship. Eq. 1 relates the rate of change of modelled CO2 to the discrepancy between modelled and observed d18O. The fact that it is the rate of change of CO2 rather than the value itself mostly serves to suppress high-frequency oscillations in the benthic d18O record, resulting in a more "smooth" CO2 reconstruction. What's more important is the fact that this rate of change is determined by the difference between modelled and observed benthic d18O. Our modelled benthic d18O value is derived from both the deep-sea temperature (which is derived from the mean annual surface temperature, which depends on modelled CO2), and from modelled global ice volume. If, for example, the modelled benthic d18O value is too low (not "glacial" enough), then the inverse routine described by Eq. 1 will slowly decrease the modelled CO2. This will increase modelled d18O first by cooling the climate, lowering the deep-sea temperature (with a prescribed time-lag, using a moving average), and also due to the increase in global ice volume resulting from that cooling. When modelled benthic d18O is in line with the observed value, the downward trend in CO2 stops and the model stabilises, until the observed d18O value changes again.

We will extend and clarify the text describing the inverse routine, so that the reader will be able to comprehend the concepts behind it without having to read our earlier publications to which the manuscript referred.

**P5 L10 – P7 L7: extended Sect. 2.1 (Methodology – Inverse modelling) so that it can be understood by readers unfamiliar with our previous work.**

Regarding the implications of our CO2 reconstruction: the reviewer is correct in stating that our model does not account for possible changes in basal conditions during the MPT. This means that, in our view, our CO2 reconstruction represents how CO2 should have evolved over time in order the produce the observed d18O record, if no changes in basal conditions occurred. If we were to repeat our simulations and prescribe more basal sliding in pre-MPT Eurasia and North America, the reconstructed CO2 would look different, likely both in terms of the glacial-interglacial amplitude, and the background trend. This is something we plan to investigate in future work, as it would provide useful context for interpreting the expected new ice-core record.

We agree that this is something that needs to be discussed in the manuscript. We will add a paragraph to the Discussion section.

**P19 L4 – P19 L15: added a paragraph to the Discussion section.**

*Related to this, the authors set out in the (very nice!) introduction that these experiments will help us to understand and quantify Earth System Sensitivity. I regard it a missed chance that they do not pick up on this issue in the Conclusions as they seem to have all data at hand to contribute to this discussion. Given that the paper in its current form is relatively short, there is enough space to elaborate on this.*

The relation between atmospheric CO2, ice sheet geometry, and global climate is not explicitly included in our model. Rather, it is a result of the model physics of HadCM3, the GCM which was used to generate the "snapshots" included in our climate matrix. The Earth System Sensitivity that would be derived from our results would therefore just be that of HadCM3. Our model mainly provides insights into

the long-term relation between CO2 and sea-level, which we believe we adequately discuss in the manuscript. The mention of earth system sensitivity in the abstract of our manuscript was inaccurate, we will change this.

**P1 L9: changed phrasing in abstract to remove inaccurate mention of "Earth System Sensitivity".**

*I am puzzled by the way the deep ocean temperature, which influences the d18O_benthic signal, is calculated. In the manuscript and in Berends et al. (2019) the authors say that they used the global average of the surface temperature anomaly. Either there is some detail missing here (some scaling) or this seems to be at odds with the measured deep ocean temperature today. Here it is important to take into account that deep ocean temperatures have a strong bias towards the sea surface temperatures at deep water formation sites, which are located in high latitudes. There is also eddy diffusive transport of heat in lower latitudes, but the low deep water temperatures clearly point to a dominating role of deep water formation. In fact, the resulting deep ocean temperature, which is caused by the balance between advective transport of cold water from deep water formation and the diffusive entrainment of heat downwards, is also dependent on the strengths of the Atlantic Meridional Overturning Circulation (Galbraith et al., GRL 2016), which is also not included in the approach by Berends et al. The value of the deep ocean temperature used in the study by Berends is not mentioned in the paper. It is likely too warm, but this is also likely compensated by the CO2 sensitivity (120 ppm/permille) with which their approach is optimized. Even if there is some scaling involved that is not described in the manuscript, using the global average surface temperature appears to be an oversimplification. Accordingly, I think it is important to discuss this issue and use some alternative sensitivity runs, where the deep ocean temperature is parameterized by high latitude temperatures and the model CO2 sensitivity is recalibrated to show that the final result is not sensitive on the choice of the deep water temperature template.*

Our modelled deep-sea temperature anomaly, which is used to calculate the modelled d18O, is derived from the northern high-latitude temperature anomaly (which is calculated as the mean temperature anomaly over the North America and Eurasia model grids), multiplied with a scaling factor (0.25 in our model) and smoothed over a 3,000 yr moving time window. This was not made clear in the manuscript; we will rectify this. The actual values this yields for deep-sea temperature changes are discussed in two of our earlier papers (Berends et al., 2018, 2019), and comparable with other studies showing a deep-sea cooling of 2 - 2.5 K during the last glacial maximum, which is agreement with proxy-based results (Shakun et al., 2015) We will add these numbers and references to the text.

**P6 L19 – P6 L21: explained the relation between surface temperature, deep-sea temperature and d18O.**

We also agree our method greatly simplifies the realistic relation between the global climate and benthic d18O. A more elaborate parameterisation based on ocean currents (which could be modelled ocean currents from the GCM snapshots, so that the relation can change over time using the same climate matrix) and global, spatially variable temperature anomalies, might be a significant improvement on this, without going so far as to run a fully isotope-enabled GCM. However, we believe that such work, while undoubtedly very interesting, lies beyond the scope of the current study. We will add a paragraph to the discussion section discussing this.

**P18 L30 – P19 L2: added a paragraph to the Discussion section.**

*abstract line 2: "...over geological time scales... "*
Added this.

*abstract line 3: "...past CO2 concentrations, thus its radiative forcing, only... "*
Added this.

*abstract line 16-17: please do not use unexplained abbreviations such as KM5c and M2 in the abstract. Please also indicate such stages after you introduced them in the main text in the figures.*
We have removed M2 and KM5c from the abstract. Since they were not mentioned anywhere else in the text, we did not add them to any figures.

*page 2 line 4: "... atmosphere from the time..."*
Changed this.

*page 2 line 6 and throughout the manuscript: Here you cite Bereiter et al. (2015) for the CO2 record, but later you cite LuÌĹthi et al., 2008. Note that in Bereiter et al., a correction of the EPICA Dome C CO2 values by LuÌĹthi et al. was introduced for ice older than about 600 kyr. You should use the corrected Bereiter et al. data throughout the manuscript.*
All figures, model-data comparisons, reported correlations, and citations have been updated to use the corrected Bereiter et al. data.

*page 2 line 9: "... have measured d11B..."*
Changed this.

*page 4 line 14: "... can help interpreting..."*
Changed this.

*page 6 line 16: Here the d18O/CO2 scaling parameter is introduced. While it has been mathematically introduced in equation 1, it would be helpful to discuss the meaning of this parameter in more detail and also discuss what it implies if this*

*parameter is assumed to be constant over time.*
The meaning of this scaling parameter is explained in the new, extended section described the inverse modelling method.

*page 8 line 14: it is not entirely clear to me what you mean by "combining the GCM snapshots according to position of the ice-sheet model in the climate matrix". I am sure there is an easier way to explain this.*
Following a suggestion from reviewer 2, a short appendix has been added where the equations of the matrix method are presented and (briefly) explained.

*page 10 line 6: "... which cover the last 500 kyr and 5.5 Myr, respectively."*
Changed this.

*page 11, line 2-4: here you say that the model study by Willeit does not include the Antarctic ice sheet, but the results look quite similar. What does this imply?*
Reviewer 2 informed us that we missed a line in Willeit et al. (2019), stating that the Antarctic sea-level contribution is assumed to be 10
*page 11 line 10: "... close to those..."*
Changed this.

*page 12 line 9: "d11B"*
Changed this.

*page 12 caption Figure 6: The references are assigned wrongly in the caption. HoÌĹnisch et al. (2009) and Bartoli et al. (2015) use d11B, not alkenones. It is correctly referenced in the main text.*
Changed this.

*page 13 line 10: You write "which they prescribed". Who is "they" in this case,*

*please provide the reference.*
"They" are Willeit et al. Added this reference to the text.

*page 13 line 19: "... different boron isotope based records is such..."*
Changed this.

*page 15 Figure 8: It would be helpful if the data younger than 800 kyr and those older than 800 kyr could be discerned in the model runs. Use different symbols or colors.*
Changed this.

*page 15 line 12: Here you say that the reconstructions by van de Wal and Willeit have a smaller spread. Do you mean in CO2 or sea level? In particular, for the reconstruction by Willeit I do not see a significantly smaller spread except that the sea level is capped.*
van de Wal and Willeit have less spread in sea level for each CO2 value (or less spread in CO2 for each sea level value), though this is indeed more obvious for van de Wal. We will clarify this in the text.

*page 16 line 21: "d11B"*
Changed this.

*page 16 line 22-24: Here you say that the data/model comparison in terms of CO2 is not conclusive, but before you showed that the highest resolution d11B data by Chalk et al., show perfect agreement with your reconstructions. I think you undersell the d11B values. Clearly each individual d11B based CO2 value has an analytical uncertainty on the order of 20 ppm, but measured in high enough resolution/replication, these data are quite useful to validate your model results.*
Chalk et al. (2017) published a collection of boron isotope proxy data around the MPT,

shown in orange in the middle panel of Figure 6 of our manuscript:

While our results (and those of the other models) agree well with data points below 280 ppmv, there is a clear mismatch in both magnitude and timing of the high interglacial values around 1150 and 1100 kyr (which probably is related to our model's poor performance for warmer-than-present climates). We therefore do not believe that the agreement between these proxy data (yellow dots) and our model results (black line) can be called "perfect". Even if we assume that the autocorrelation of the individual proxy data points allows us to assume a lower measurement uncertainty, the difference between the median of the proxy data and the multi-model mean is much larger than the differences between the different models. This justifies our assertion that these proxy data cannot be used to choose one model reconstruction over the others.

We will clarify this line of reasoning in the manuscript.

**P18 L15 – P18 L28: added a paragraph to the Discussion section.**

*page 16 line 32: "... for a colder-than-present..."*
Changed this.

*page 17 line 4: The last sentence is weak and does not give credit to the work performed in this study. The authors should elaborate much more on this, as outlined in my general comments above.*
We have added two paragraphs to the Discussion section, discussing the various sources of uncertainty in our results, and how we believe our reconstruction should be interpreted.

[Figure]

**Fig. 1.**

---

## Author Comment (AC2) · 7 Jul 2020

Rebuttal to the review by Andrey Ganopolski

We thank the reviewer for their comments on the manuscript and would hereby like to address the concerns they raised.
Comments in italics, below our rebuttal. Page and line numbers refer to the revised manuscript.

*Method description. One of the problems for the readers of this manuscript is*

*that the method used in this study has been developed over a long period time and its comprehensive description are scattered among a number of previous publications. Even although I was familiar with some of them, it took me a lot of time to get a more or less clear understanding of what authors are doing. Of course, one cannot expect such efforts from a typical reader. However, without a proper understanding of the method, the results presented in the manuscript are not very useful. This is why, I would suggest to make a more detailed description (including the key equations) in the appendix or supplementary information. In particular, I am curious how the effect of orbital forcing has been accounted for by the "matrix method".*

We agree that a thorough understanding of our methodology relied too much on concepts explained in earlier publications. We will revise and extend the sections describing both the inverse modelling routine and the matrix method.

Orbital forcing is included in our model in two separate ways. First, it is included as a term in the calculation of surface melt in the mass balance module (the insolation-temperature model). Second, it is included in the matrix method in the interpolation equation. The matrix method contains two separate interpolation equations that are used to combine the GCM snapshots; one for temperature, and one for precipitation. The one for precipitation is based on ice-sheet geometry, to account for orographic forcing of precipitation. The interpolation routine for temperature is based on "absorbed insolation": the product of insolation at the top of the atmosphere, and (1 – albedo). This accounts for both the changes in albedo resulting from ice-sheet advance and retreat, and the changes in insolation resulting from orbital forcing. While an extension of the climate matrix that includes GCM snapshots that have been calculated for different orbital configurations (so that the direct effect of insolation changes on surface temperature is also include) is planned, it is not included in the model version described here.

We will make sure that this is properly explained in the Methodology section.

**P5 L10 – P6 L21: extended and clarified Sect. 2.1 (Methodology – Inverse modelling)**
**P19 L19 – P21 L8: added Appendix A, which presents and (briefly) explains the equations governing the matrix method**

*The model validation is based on the comparison of reconstructed $CO_2$ over the past 800 kyr with the ice core data. The authors compare the results of their current study with several others and conclude that they are the best. However, it is obvious that comparison results of inverse modelling with forward modelling presented in Willeit et al (2019) is the same as comparison of apples with cucumbers. The inverse model is forced by benthic d18O which is already highly correlated with $CO_2$ (correlation coefficient is 0.86). The authors should make this point very clear. The only surprising thing in this table is the extremely poor performance of Stap et al. (2017). Unfortunately, the authors themselves admit on page 15 that they cannot explain this fact. In fact, it is much more instructive to compare the result of a rather complex inverse modelling approach used by the authors to a simple linear regression of d18O from LR04 stack. Surprisingly (or maybe not) this simple "model" outperforms Berends et al. Indeed, it has R2=0.71 (versus 0.68 in Berends et al.) and rms=13.8 ppm (vs. 15.3) for "simulated" $CO_2$ concentration over the last 800 kyr. After such a comparison, the numbers in Table 1 do not look very impressive. For the rest of Quaternary, results of Berends et al. also do not differ much from this simple regression model. After all, it is rather expectable (and have been demonstrated by Willeit et al., 2019) that $CO_2$ also followed ice volume variations during 41-kyr world but with a smaller amplitude.*

We apologise to the reviewer if our manuscript seemed to suggest that our own method is "best"; we agree that there is no meaningful way to declare one modelling approach "better" than another. We also agree that the results of the simple linear
regression proposed by the reviewer should be included in the comparison. Indeed, this helps to illustrate what we believe is the main conclusion from this comparison; more complex models, including more elaborate physics and describing more components of the Earth system, are useful for studying large-scale relations between these components, but are not necessarily better at resolving the evolution of a single component or parameter. We will clarify this in the text.

**P13 L24 – P13 L31: added a simple linear regression to the statistical comparison**

We will also follow a suggestion from Matteo Willeit (the main author of Willeit et al., 2019, who contacted us shortly after the discussion version of our manuscript was published, with a similar question about the comparison of correlations to the ice core record), to include an optimised time lag for each model $CO_2$ reconstruction before calculating the correlations with the ice core record. This mainly affects the results of Stap et al. (2017) and Willeit et al. (2019), where the coefficients of determination $R^2$ for both studies increase from  0.25 to  0.45. While this extra step does not alter the conclusions we draw from this comparison, it does more properly give credit to the results of the different studies.

Following a comment by Anonymous Reviewer 1, we have also updated all figures and numbers to use the more recent ice-core $CO_2$ record by Bereiter et al. (2015), rather than Lüthi et al. (2008).

*The real question is what was $CO_2$ concentration at the end of Pliocene.  And here I see a real problem with the results presented in Berends et al.  Indeed, if during the entire Pleistocene, $CO_2$, ice volume and d18O variations were essentially identical, during the late Pliocene $CO_2$ get really wild. Figure 4 shows several $CO_2$ oscillations with the amplitude above 100 ppm. Of course, this is not 200+ ppm as in Stap et al (2017) but still a lot. As the scientist who has been heavily involved in*

*explaining glacial-interglacial CO2 variability, I must confess that it is extremely difficult to explain 80 ppm change in CO2 concentration even for the full glacial cycles of the late Quaternary. What could cause even larger Pliocene variations in CO2 without any obvious external forcing, the authors do not explain. This is why I strongly suspect that the reason for such weird behaviour of CO2 before Pliocene-Pleistocene transition is that the inverse modelling of CO2 concentration based on benthic d18O beyond Quaternary represents an ill-posed problem.*

We agree that our results for the late Pliocene are by no means the definitive answer to the question of how the world looked like in terms of CO2, global climate and ice sheet geometry. In our view, the main problem here is the relatively large changes in benthic d18O. Explaining these requires either very large changes in deep-sea temperature, moderately large changes in global ice volume, or (more likely) a mix of both. Our model results tend towards the "temperature" end of this spectrum, resulting in large changes in CO2: almost 100 ppmv difference between the coldest point of the Pliocene during M2, 3.3 Myr ago, and the warmest point during KM5c, 3.205 Myr ago (for our default simulation; in the low-CO2 end member, this difference reduces to about 85 ppmv).

We suspect that the relative sparsity of our climate matrix for warm worlds might result in a bias towards lager ice sheets in warm climates. This means that increasing modelled CO2 above present-day levels does not cause as much ice-sheet retreat as it maybe should, so that benthic d18O does not decrease so much. In order to reproduce the observed d18O record, the inverse routine will compensate by increasing CO2 until the resulting deep-sea temperature change is enough to produce the required change in d18O. This also explains the very large uncertainty range resulting from our sensitivity analysis; an additional change of 0.1 per mille in d18O, for constant ice volume, requires a very large change in deep-sea temperature, mean annual surface temperature, and CO2. Changing our climate matrix such that warm climates will lead

to more ice-sheet retreat will essentially shift the blame for the high d18O variability from the temperature end of the spectrum towards the ice volume end; it will reduce the modelled CO2 variability during the late Pliocene, but it will also increase the sea-level high stands.

Lastly, looking at proxy-based reconstruction, the boron isotope data by Martínez-Botí et al. (2015), which has both the highest temporal resolution and longest temporal range of all available proxies, shows a variability during the late Pliocene of about 150 ppmv (albeit with an uncertainty of about 100 ppmv in either direction). While certainly not definitive, especially considering the large discrepancies between difference boron-based reconstructions (as also discussed in our manuscript), their data does seem to suggest strong CO2 variability in warmer-than-present worlds.

We will extend the Discussion section of the manuscript to reflect these thoughts.

**P18 L15 – P18 L28: added a paragraph to the Discussion section, discussing the variability and uncertainty in our CO2 reconstruction during the late Pliocene.**

*The authors wrote on page 9 that "uncertainties are conservative in this study". What the authors mean under "conservative" is not clear to me. To me, the estimate of uncertainties in this study is overoptimistic at best. Even if the maximum error in benthic d18O is indeed only 0.1 promile, the methodology has a number of other uncertainties related both to forward model and to conversion between climate charac- teristics (ice volume, temperature) and d18O. For the large glacial cycles of Quaternary even a larger uncertainty still does not prevent a reasonable estimate of CO2 but the situation is very different prior to 2.7 Ma. Before Quaternary, the model "assumes" very little variability in global ice volume and thus most of d18O variability has to be attributed to CO2 change and this is precisely what the model does. However, in this case, even uncertainty of +-0.1 promile already constitutes a serious problem. Indeed,*

*0.2 promile correspond to about 1C change in the deep-water temperature which in turn corresponds to 1.5C in global air temperature. The later number corresponds to change of CO2 (assuming climate sensitivity =3C) from 280 to 400 ppm. Thus, even with a very optimistic estimate of the method uncertainty, for pre-Quaternary climates this method cannot distinguish between a possibility that CO2 was as low as the preindustrial one or that it was as high as the current one. Obviously, such "reconstruction" is not very helpful.*

The statement that "uncertainties are conservative" was intended to indicate that the uncertainties we report are only those that arise from the sensitivity analysis described in the manuscript. This is simply the sensitivity of the model to uncertainties in the d18O record, which we showed in our 2019 publication to be larger than the sensitivity to uncertainties in other model parameters. We agree with the reviewer that this is not at all the same as the real uncertainty in our results; there are many other factors introducing uncertainties that cannot be quantified through such sensitivity analysis, and these are likely to be larger still than the numbers we report.

We will extend the Discussion section of the manuscript to reflect this, especially in relation to the previous comment about the difficulty of interpreting the d18O record in the late Pliocene.

**P18 L15 – P18 L28: added a paragraph to the Discussion section, discussing the variability and uncertainty in our CO2 reconstruction during the late Pliocene.**

*"80/120 kyr cycles". Although this is not very essential for the manuscript under consideration, but the authors used the expression "80/120 kyr cycles" (actually it should be 82/123) several times in this and previous papers which provokes me to make the following comment: The durability of "two or three obliquity cycles" myths is amazing since it is not supported by real data! Glacial cycles of the late Quaternary*

*have average periodicity close to 100 kyr which explains strong 100 kyr peak in the frequency spectra of ice volume. It is true that the durations of individual glacial cycles deviate significantly from 100 kyr but they also do not cluster around 80 and 120 kyr (see for example Table 1 in Konijnendijk et al., 2015). In fact, durations of individual glacial cycles are relatively uniformly distributed between 80 and 120 kyr with half of the cycles been closer to 100 kyr than to 80 or 120 kyr.*

While we believe that declaring the 80/120 kyr hypothesis to be a "myth" is overly dismissive of the studies supporting this hypothesis (especially when considering the difficulties in constructing insolation-independent age models, described by Huybers and Wunsch, 2004), we agree with the reviewer that it is important to mention the ongoing discussion about the nature of the late Pleistocene glacial cycles. We will clarify this in the manuscript.

**P14 L11 – P14 L17: added a few lines about the 100 vs 80-120 kyr discussion.**

*P.3, L.8 "proxies for global mean temperature"? Greenland and Antarctic records present proxies only for local temperatures which differ significantly from the global one*
Changed this.

*P3., L.10. "In that case ocean water temperature can be resolved as closure term from the benthic signal" This is not clear*
Changed this.

*P. 4, L.9 The definition of "entire climate system (atmosphere, ocean, cryosphere, carbon cycle, etc.)" is not consistent with contemporary terminology. Such system is named Earth system and Earth system models describe not only "physical processes"*

*(L. 10).*
Changed this.

*p. 4, L. 21 "the known relations between atmospheric CO2, global temperature and climate, and ice-sheet evolution". Why authors think that these relations are "known". Even the relation between CO2 and global temperature is still not well-known.*
We agree that our phrasing was unclear. We will clarify this in the manuscript.

*p. 5. I am not sure I understand why the authors put "data" and "model" in quotes.*
These words are put in quotes because they constitute rather informal, but also obvious, descriptions of the aim and general approach of the studies we describe. While it is not possible (or desirable!) to draw a clear line between data studies and model studies, many publications about paleoclimate either rely mostly on the presentation and interpretation of proxy data, or on the development and application of modelling methods. We feel that it is important to explain the distinction between them, and how our work relates to both.

*P.7 L. 11 "The reconstruction by Laskar et al. (2004) is used to prescribe time- and latitude-dependent insolation". Insolation is not reconstructed by computed using physical laws. This is why orbital forcing can be calculated for the past and future with the same (very high) accuracy.*
We will replace "reconstruction" by "solution", in line with the phrasing by the authors of Laskar et al. (2004).

*P. 11, L. 2. "so any possible contribution from Antarctica to changes in sea-level ... is not accounted for in their reconstruction". This is an incorrect statement. It is written in Willeit at al. (page 6) "Sea level is computed from the volume of modeled NH*

*ice sheets assuming an additional 10 percent contribution from Antarctica".*
We will correct this in manuscript.

*P. 15, L. 15. "... show a CO2 "threshold" for glaciation and sea-level drop around 250
ppmv". Our studies (e.g Ganopolski et al., 2016) do not support the existence of a
single CO2 threshold for glaciations. To the contrary, we found that glacial inception is
determined by a combination of insolation and logarithm of CO2 concentration.*
We will correct this in manuscript.

*Fig. 4. It is not explained what shading shows in this figure.*
Shaded areas indicates the uncertainty in the LR04 benthic d18O stack, and the
resulting uncertainty in the reconstructed CO2 and sea level. We will clarify this in the
manuscript.

*The reference Stap et al. (2017) is not in the reference list.*
Added this reference.

―――――――――――――――――――――

---

## Author Response (AR2)

**Rebuttal to the review by Andrey Ganopolski**

We thank the reviewer for their comments on the manuscript and would hereby like to address the concerns they raised. Comments in italics, below our rebuttal. Page and line numbers refer to the revised manuscript.

*I appreciate the efforts which the authors made to respond to my criticism and suggestions and in general I am satisfied with their response. The only my major remaining concern I still have is the treatment of the uncertainties of CO2 reconstructions. In response to my comment, the authors explained that under "conservative" estimate of uncertainties they understand the estimate only of one but the main source of uncertainties. If so, the authors believe*

*that their errors in CO2 during the entire Quaternary are not higher than ca 5 ppm and thus they have nearly 100% confidence that during the early Quaternary CO2 never exceeded 300 ppm. However, some reconstructions, including the cited in the manuscript Martínez-Botí et al. (2015), suggest that at the beginning of Quaternary, CO2 most of time was well above 300 ppm (somewhere between 300 and 400 ppm). Needless to say, this is an important difference. I do not expect from the authors to perform "objective" (instead of "conservative") estimate of their method*

*uncertainties because this is impossible without using a comprehensive water-isotope enabled Earth system model. However, it would be useful to inform potential readers that reported uncertainty is nothing more than the method sensitivity to 0.1 promille change in the benthic d18O (which is likely to be over-optimistic estimate for LR04 stack uncertainties, at least, for the early Quaternary) while the real errors of the method can be much larger but, unfortunately, cannot be properly evaluated at this stage.*

We agree that it is important to explain to a reader that the error margins reported in our figures and tables are not a reflection of the "true" uncertainty of our reconstruction – which is, of course, much larger, but also difficult to quantify. We will make sure this is made clear in the manuscript.

**P9, L24-30 (Sect. 3: Experimental set-up and results): added a few lines motivating our choice to investigate only the sensitivity to the prescribed forcing, and how the resulting margin of error relates to the "true" uncertainty in our results.**
**P19, L6-9 (Sect. 5: Conclusions and discussion): added a few lines reiterating the difference between the reported margins of errors and the (unquantifiable) true uncertainty in our reconstructions.**

*p. 31, l. 12. I do not understand why "a narrower range of sea levels for each CO2 indicating that their ice sheets respond faster to change in climate" (whatever faster means). The close relationship between CO2 and sea level seen also in the data only indicates the existence of strong climate-ice sheets-carbon cycle feedbacks but tells nothing about the rate of ice sheet response to climate change.*

In general, the temporal response of an ice-sheet model to a change in prescribed climate will depend on the type of model. A simple linear regression between CO2 and sea-level describes an instantaneous response, which would give a scatterplot as in Fig. 8 showing a single line, without any spread; a one-to-one relation. An SIA ice model (such as the one used by van de Wal et al., 2011) will have a response time ranging between hundreds to thousands of years, depending (among other things) on the way the ice flow factor and basal sliding are described. This means that the relation between CO2 and sea level at a specific point in time is no longer one-to-one, which results in a slight spread around the curve in Fig. 8. Many other physical processes will give rise to additional time lags (e.g. englacial thermodynamics, carbon cycle changes related to oceanic cycling of carbon, glacial-isostatic adjustment, etc.), which will increase the spread in the scatterplots.

The aim of the paragraph the reviewer quotes from is to use this fact to try and interpret some of the observed differences between the different models. We agree that this needs some clarification in the manuscript.

**P15, L16-22: Added a few sentences explaining the interpretation of the slope and spread of the scatter plots in Fig. 8.**

*P. 31, l.13. I cannot see on this figure "a relatively narrow range of sea level in the 180-250 ppm CO2 range" in Willeit et al. (2019). The significance of this fact is also unclear to me.*

This, too, has to do with time lag. In Fig. 8, both de Boer et al. (2014) and our study show one or two "trails" of dots at the left-hand end of the scatter plot, corresponding to some of the really cold Late Pleistocene glacial cycles (though indeed that can't be seen from the graphs, as they don't show the time stamps of individual data points. We attempted using a color scale for that, but that made the graphs too chaotic to be useful). There, modelled ice-sheet retreat lags the sudden increase in d18O (and therefore CO2) immediately after a glacial maximum, such that the dots in the graph move to the right (increasing CO2) before they start moving up (rising sea level). Compared to de Boer 2014 and our study, the left-hand end of the scatter plot for Willeit 2019 is a bit narrower, indicating that their ice sheets start retreating a bit sooner after CO2 starts rising. However, we agree that the difference is very small, and mentioning it without investigating the cause of this difference is not useful for the manuscript. We will remove this sentence.

**P16, L14: removed this sentence.**

*p. 32, l. 1. It is true that in our previous studies (but not in Willeit at al., 2019) we have shown that the threshold for glacial inception depends on both insolation and CO2 and we have shown in Ganopolski et al. (2016) that the true CO2 threshold for glaciation is ca. 350 ppm because for higher CO2 glacial inception would be possible only with such low summer insolation which cannot be achieved with the realistic orbital parameters. This funding not only explains fig. 8f but is also consistent with the data (fig. 8a). However, during most of Quaternary, the interglacial*

*CO2 level was in the range 240-280 ppm and this is why it is natural that any appreciable sea level drop can only be observed when CO2 was below these values and this is true for all figures in fig. 8 except for Stap et al. (2017). In short, the fact that most of sea level drop during Quaternary occurred for CO2 below 250 ppm does not imply that ice sheets cannot grow under significantly higher CO2 level.*

We agree that the theory of glacial inceptions (and terminations!) is more complicated than how it is discussed in our manuscript. Indeed, as both Willeit (2019) and the actual observational data indicate the presence of medium-sized ice sheets under higher CO2 concentrations, this implies that the relation between CO2, insolation and SMB in the models by van de Wal et al. (2011), de Boer et al. (2014) and our study are too simplistic in this regard. We will add a sentence to the manuscript to clarify this.

**P16L16 – P17L3: clarified this.**

*p. 34. l.1 It is not clear to me what "downwelling regions" means*

In this context, "downwelling regions" are those areas of the oceans were downwelling occurs, and deep ocean water is formed. When Bintanja and van de Wal (2008) created the first inverse-modelling set-up, the idea was that the LR04 benthic d18O stack, while claiming to be made up of a "globally distributed" set of records, is predominantly made up of cores from the Atlantic. The contribution from deep-water temperature to the d18O signal would then be specifically from Atlantic deep-water temperature, which would be related mostly to surface temperatures in the North Atlantic (the downwelling area where this deep water is formed). They therefore related the deep-water temperature d18O component directly to surface temperatures in the Northern hemisphere, which was convenient because this also governed the surface mass balance of the North American and Eurasian ice sheets. Since then, the inverse modelling method has been significantly expanded, particularly by using global climate from a GCM instead of a simple globally uniform temperature offset, such that this parameterisation is now probably too simplistic to be justifiable. We will clarify this in the manuscript.

**P18L28 – P18L34: clarified this.**

**Rebuttal to the comments by the editor, Ed Brook**

We thank the editor for their comments on the manuscript and would hereby like to address the concerns they raised. Comments in italics, below our rebuttal. Page and line numbers refer to the revised manuscript.

*Referee 2 has some concerns about the way uncertainties are described and I also puzzled about this. As I understand it you consider a 0.1 per mil uncertainty in the benthic isotope record only, and that is propagated through the analysis. This is described as "conservative". I think a reader may be confused about whether this means that the uncertainties are likely overestimated or underestimated. Could you clarify this?*

We have adapted the manuscript to address the concerns the reviewer raised about the reported uncertainties (see above).

*Also, I think it would be helpful to describe in more detail why you get some large uncertainties in CO2 in warm climates.*

We have further clarified the paragraph of the Discussion section dedicated to this issue.
**P18, L20-25: clarified this.**

*One other thing, I am wondering if it would be useful for you to cite the ice core mean ocean temperature*
*reconstructions for the LGM (Bereiter et al. 2018, Nature, v. 553).*

This is certainly an interesting publication. The number they quote for the glacial-interglacial difference in mean ocean temperature of 2.57 +- 0.24 K is reasonably close to the number we found in our earlier paper about our modelling approach (~2.1 K in Berends et al. 2019). However, our current study does not look into ocean temperatures (we are planning to revisit
that in the near future, with a substantial update to our matrix method which we're working on). Adding a section on ocean temperatures would pose a substantial revision of the current manuscript, which we feel would not affect our conclusions.

[revised manuscript text omitted]